# Using ERIC to Assess Implementation Science in Drowning Prevention Interventions in High-Income Countries: A Systematic Review

**DOI:** 10.3390/ijerph21010045

**Published:** 2023-12-27

**Authors:** Malena Della Bona, Gemma Crawford, Brooklyn Royce, Jonine Jancey, Justine E. Leavy

**Affiliations:** Collaboration for Evidence, Research and Impact in Public Health (CERIPH), School of Population Health, Curtin University, Perth 6000, Australia

**Keywords:** systematic review, implementation science, implementation strategies, Expert Recommendation for Implementing Change (ERIC), drowning, downing prevention interventions, high-income countries

## Abstract

This systematic review identifies and describes the use of the Expert Recommendation for Implementing Change (ERIC) concepts and strategies using public health approaches to drowning prevention interventions as a case study. International calls for action have identified the need to better understand the implementation of drowning prevention interventions so that intervention design and implementation is improved. In high-income countries (HICs), interventions are sophisticated but still little is known or written about their implementation. The review was registered on PROSPERO (number CRD42022347789) and followed the PRISMA guidelines. Eight databases were searched. Articles were assessed using the Public Health Ontario Meta-tool for quality appraisal of public health evidence. Forty-nine articles were included. Where ERIC strategies were reported, the focus was on evaluative and iterative strategies, developing partnerships and engaging the target group. The review identified few articles that discussed intervention development and implementation sufficiently for strategies to be replicated. Findings will inform further research into the use and measurement of implementation strategies by practitioners and researchers undertaking work in drowning prevention in HICs and supports a call to action for better documentation of implementation in public health interventions.

## 1. Introduction

The generation and use of knowledge is critical for evidence-informed public health practice. There is recognition of the need to address the challenges that hinder closing the “know–do” gap in public health implementation [1,2]. Implementation science seeks to address these barriers using methods and strategies that facilitate the uptake of evidence-based practice and research into regular use by practitioners and policymakers [3]. There are several considerations inherent in this sentiment. When interventions are assessed as being effective, if others do not utilise them, they will not become widely accepted [4]. Intervention implementation requires active consideration to ensure that programs are deployed with and into communities more efficiently and effectively. 

A range of frameworks and tools have been developed to guide implementation, for example, the Consolidated Framework for Implementation Research (CFIR) [5] and Theoretical Domains Framework [6]. Common to these frameworks is the use of domains and constructs to explain and measure implementation and describe the interplay of the intervention content and context to effect behaviour change. However, there are few frameworks specific to public health [1,7], despite growing interest in the use and impact of implementation science for public health interventions [7,8,9,10]. The Expert Recommendations for Implementing Change (ERIC) project identified nine concepts and 73 strategies that aid in the development [11] of implementation strategies and can serve as a tool to assess the strategies used in implementation. This tool was initially designed for a clinical setting [11,12]; it has been refined and used to reflect public health practice within a community setting [7]. There are examples of implementation science being used in some health and community settings [7,11,13] across mental health [4,11,12,14,15] and substance use [16]. However, in other public health areas, such as drowning prevention, which is the case study for this review, the application of implementation science is limited [7]. 

Drowning is a leading cause of unintentional injury [17,18], resulting in over 2.5 million deaths worldwide over the last decade [19]. Drowning is a complex global public health issue with different drivers across low-, middle- and high-income contexts [20,21,22,23]. In HICs, drowning events tend to occur during recreational activities [17,24] around the water and around the home [25]. In low- and middle-income countries (LMICs), there is a higher prevalence of children drowning close to home due to issues of supervision, barriers to water sources and water safety skills, while older children and adults drown when undertaking work or during travel on water [21,26]. While there is evidence of community-led interventions [22,23] and institutional guidance around effective strategies to reduce drowning [27], the literature also highlights gaps in the quality, consistency and reporting of programs [22,23,28]. There are also differences in the number of drowning prevention-related peer-reviewed publications published [22,23] and the types of interventions [29] relevant for LMICs compared to HICs. Of the limited published interventions from LMICs, most were delivered by agencies without the capacity to include large-scale evaluation or knowledge translation [17,23]; thus, implementation in the two settings (HIC and LMIC) is not directly applicable [29].

Consequently, there have been international calls for action to establish processes to better understand the implementation of drowning prevention interventions in community settings [19,26,30] and several resources have been developed to aid in the implementation of drowning prevention interventions [31,32,33]. However, these resources have concentrated on the activities of interventions (e.g., the WHO implementation guide focusses on ten evidence-based interventions and strategies [31]) rather than the process of implementation and tend to be conducted in LMICs. In HICs, there is a growing focus on the use of evidence to strengthen the design, delivery and evaluation of interventions [23], with calls for practitioners and researchers to learn from each other [26]. Recent advancements in the publication of more sophisticated approaches to the development of drowning prevention interventions [23] lend themselves to further exploration of context of intervention implementation. 

This systematic review aims to identify drowning prevention studies undertaken in HICs as a public health case study of implementation science and describe and assess the use of implementation science ERIC concepts and strategies. The review asks: 

What are the implementation strategies used in drowning prevention interventions in high-income countries and how are they described?What are the gaps in the use and reporting of implementation strategies in drowning prevention interventions in high-income countries?

## 2. Materials and Methods

This systematic review was conducted according to the Preferred Reporting Items for Systematic Reviews and Meta-Analyses (PRISMA) guidelines [34] and registered with the International Prospective Register of Systematic Reviews (PROSPERO) (ID CRD42022347789) [35]. The search strategy for included articles and the PRISMA checklists for abstracts and manuscripts were completed (Appendix A, respectively).

### 2.1. Criteria for Inclusion

Included articles were primary studies, addressed drowning prevention interventions in HICs, were written in English and were published between 2002 and 2022 (see Table 1). HICs are defined as economies with a gross national income (GNI) of USD12,696 or more per capita [36]. Exclusion criteria included articles that focus on clinical aspects of drowning events; specific target groups or locations not relevant to the wider population (i.e., those with water phobia, working on specific construction sites); simulation studies (i.e., studies of cardio-pulmonary resuscitation (CPR) or rescue delivery in specific settings) or other injury prevention (i.e., not solely drowning prevention). Articles addressing the development of recommendations and/or guidelines, where no intervention was described, were also excluded.

### 2.2. Search Strategy

In consultation with the university health sciences faculty librarian, the initial literature search strategy was developed using medical subject heading (MESH) and text words related to implementation and drowning prevention. Once an initial Medline search was completed, the strategy was adapted to the subject headings of seven additional databases (see Appendix A for full search strategy): PubMed, PsycINFO, ProQuest, Scopus, Web of Science, Global health and SPORTDiscus. 

### 2.3. Screening and Quality Appraisal

The initial database search identified 3547 articles. All citations from the initial search were imported into Endnote 20 [37] referencing management software. Using the Endnote and Rayyan software [38] “find duplicates” tool, 1539 articles were removed. Two reviewers (MDB and BR) individually screened the identified articles by title and abstract using Rayyan [38] to determine the relevance of the remaining articles (*n* = 1515). Articles were categorized as “possibly relevant”, “maybe” (where the reviewer was unsure if the article met the criteria) and “excluded”. The systematic review PRISMA 2020 flow diagram [34] is presented in Figure 1.

Articles identified as “maybe” during the title and abstract search were re-classified through discussion amongst two research team members (MDB and BR) and. where there was disagreement, with a third member of the team (JL). The full text articles identified as “possibly relevant” were retrieved and assessed by one reviewer (MDB) using a standardized exclusion list. A second researcher (BR) randomly cross-checked 10% of articles to identify any selection anomalies. Any full-text articles classified as “maybe” were resolved with a co-author (GC). To ensure coverage, the reference lists of “possible inclusion” articles (*n* = 104) were hand-searched for any articles not previously identified; 22 further articles were identified (see Figure 1). 

All included articles (*n* = 49) were assessed for quality using the Public Health Ontario Meta-tool for quality appraisal of public health evidence (PHO META QAT) [39] by one reviewer (BR). Another reviewer (MDB) randomly cross-checked 10% of the appraisals to ensure accuracy. Articles were analysed and scored using four domains (relevance, reliability, validity and applicability) containing nine categories [40] and were scored as met the criteria = 2, not sure/unclear = 1 and did not meet the criteria = 0. As in previous reviews [23], articles were categorized on their summed score, where ≤9 = low quality (*n* = 11), 10–14 = medium quality (*n* = 17) and 15–18 = high quality (*n* = 21). All 49 articles met the criteria for inclusion based on the quality appraisal. 

### 2.4. Outcome Measure (Implementation)

Included drowning prevention intervention articles were assessed for inclusion using the refined Expert Recommendations of Implementing Change (ERIC) project concepts and strategies for use in community settings [7,11,41]. Refinement was made to the criteria by the first author (MDB). Changes are described in Appendix A. In summary, refinements included terminology changes to reflect drowning prevention “interventions” in place of “clinical innovations” [11], “target group and support networks” in place of “patient/consumers and families” [11] or “priority populations” [7], and “providers” in place of “clinicians” [11]. These modifications were verified with a practitioner with over 10 years’ experience in the drowning prevention sector. The final list of nine ERIC concepts and 73 strategies are described in Table 2. The nine concepts include: c1 use evaluative and iterative strategies, c2 provide iterative assistance, c3 adapt and tailor context, c4 develop partner relationships, c5 train and educate stakeholders, c6 support providers, c7 engage target group, c8 financial strategies and c9 change infrastructure.

### 2.5. Data Extraction and Synthesis

The following data were extracted from the drowning prevention articles:Author; year; aim; location; sample (*n*); recruitment; response rate.Intervention: level (i.e., who the intervention involved—individual, group, population); type (behavioral—actions taken by individuals to prevent drowning [42]; socio-ecological—social, physical, policy or environmental; or mixed (behavioral and socio-ecological) [43]); activity; duration; use of theory (theory); formative research.Evaluation: design; measures; human research ethical approval (HREC); findings.Implementation: level (i.e., systemic, organizational, provider, target group); concepts [41]; strategies [7,11,44] (used and recommended).

## 3. Results

A total of 49 peer-reviewed articles were included [45,46,47,48,49,50,51,52,53,54,55,56,57,58,59,60,61,62,63,64,65,66,67,68,69,70,71,72,73,74,75,76,77,78,79,80,81,82,83,84,85,86,87,88,89,90,91,92,93]. Table 3 presents the extracted data. 

### 3.1. Overview of Case Studies (Intervention and Evaluation)

Twenty articles were from Australia (*n* = 20, 40.8%), 19 (38.8%) from the United States of America (USA), and the remaining articles were from Canada (*n* = 3, 6.1%), New Zealand (*n* = 3, 6.1%), the United Kingdom (*n* = 1, 2.0%), Greece (*n* = 1, 2.0%), Uruguay (*n* = 1, 2.0%) and Italy (*n* = 1, 2.0%). Most articles (*n* = 42, 85.7%) were published from 2012 onwards, with half of those (*n* = 22) published in the last five years (2017–2022). Ethical approval was reported in 32 articles (65.3%), with the remaining not recorded (*n* = 16, 32.6%) or identified as not required (*n* = 1, 2.0%) due to the clinical environment of the data collection. Participation ranged in size, with samples taken from clinical settings (*n* = 7) such as patients’ attending physicians’ rooms to state-wide initiatives (*n* = 14,842) and ranged in duration from one-week to 20-year interventions.

Most interventions (*n* = 48, 98.0%) were delivered at the population or group level, with only one at the individual level. Intervention types were mostly behavioral (*n* = 31, 63.3%), with the remainder identified as a combination of both behavioral and socio-ecological (*n* = 10, 20.4%) or socio-ecological only (*n* = 8, 16.3%). Study activities included pool-based swimming or water safety lessons (*n* = 12, 24.5%), training for parents (*n* = 12, 24.5%), beach safety education (*n* = 5, 10.2%), cardiopulmonary resuscitation first aid or bystander training (*n* = 5, 10.2%). Other activities included lifejacket use (*n* = 4, 8.2%) and pool fencing (*n* = 2, 4.1%). Twenty-seven articles did not identify the use of theory (55.1%). The uses of theory most identified were the Health Belief Model (*n* = 6) and Theory of Planned Behaviour (*n* = 4). The remaining articles identified educational, behavioral and social marketing theories.

The evaluation design methodology included observational studies (*n* = 7, 14.2%), quantitative (*n* = 27, 55.1%), qualitative (*n* = 1, 2.0%) and mixed methods (*n* = 9, 18.4%). In addition, four articles (8.1%) included process evaluation only and one educational article did not discuss evaluation. The most frequently recorded measures included knowledge (*n* = 20), attitudes and beliefs (*n* = 14), risk perception (*n* = 11), water-based activities undertaken (*n* = 10), behavioral intent (*n* = 9) and intervention awareness (*n* = 9). In observational studies, the main measure recorded was swim skills (*n* = 6) or personal flotation device use (*n* = 2).

### 3.2. Implementation Key Concepts and Strategies

Table 4 summarizes the identified ERIC concepts and strategies. Throughout the tables and text, the concepts and strategies have been identified in brackets, where “cX” identifies the concept and the underscore number “_XX” identifies the strategy. All nine concepts were identified across the 49 articles, with *developing partner relationships (c4)* (*n* = 32, 65.3%), *evaluative and iterative strategies (c1)* (*n* = 29, 59.2%) and *engaging with the target group (c7)* (*n* = 22, 44.9%) the most common concepts. Four articles reported the concept *change infrastructure (c9)* (8.2%). One included study [76] did not identify any implementation strategies.

Forty-three ERIC strategies (58.9%) were identified. The most frequently identified strategies were *develop academic partnerships (c4_04)* (*n* = 18, 36.7%), *promote network collaboration (c4_13)* (*n* = 14, 28.6%), *intervene with the target group to enhance uptake and adherence (c7_02)* (*n* = 14, 28.6%) and *conducting local needs assessment (c1_04)* (*n* = 11, 22.4%). Twenty-five (34.2%) ERIC strategies were identified four times or fewer (see Table 4).

Concepts and strategies identified in the intervention articles are detailed below, in order of frequency.

#### 3.2.1. Developing Partner Relationships (c4)

Partnerships were developed with a range of groups, including academic institutions (*n* = 18) [46,50,51,52,53,54,60,61,64,67,75,77,78,85,86,87,88,89], community groups (*n* = 5) [66,73,83,86,89], local councils (*n* = 2) [72,91] and public health organizations (*n* = 1) [83]. For example, Mitchell and Haddrill [73] worked closely with local groups to implement the SafeWaters campaign for the Chinese community in New South Wales (NSW), Australia. Quan and colleagues worked with the Vietnamese community in Washington, USA to develop and evaluate an intervention to combat a spate of local drownings in the community. Van Weerdenburg and colleagues [49] identified the importance of local council support for implementing pool fencing inspection programs.

##### Develop Academic Partnerships (c4_04)

Partnerships with university academics (*n* = 18) were identified as human research ethics committee (HREC) approvals and research–practitioner collaborations. More than half the examples of *developing academic partnerships* (55.6%) involved the use of university or research group ethics boards [46,51,53,75,77,85,86,87,88,89], whilst almost two-thirds of articles (*n* = 31, 63.3%) indicated they had received ethical approval. HREC approval within the strategy *develop academic partnerships* were recorded if it was clear where the ethics approval was obtained, with an approval number included. For example, an intervention designed by Koon and colleagues [64] was undertaken by the Lake Macquarie City Council and the University of New South Wales Beach Safety Research Group, obtained HREC approval and aided in developing tools for the intervention. In another study, Hamilton and colleagues [61] reported the development of resources in partnership with university academics.

##### Promote Network Collaboration (c4_13)

Network collaborations involved identifying and expanding existing networks within and outside the lead organization to promote information sharing, collaborative problem solving and a shared vision/goal for implementing the intervention [11]. The collaboration took the form of involvement of organization staff with expertise from a variety of backgrounds [48,79,84,90,92], working with not-for profit organizations [59,83], development of multi-sectorial partnerships [49,83,89], expert input into the campaign [73,94] and the development of evaluation tools [59,90]. For example, Quan, Shepard and Bennett [83] worked with community groups, the parks department and public health organizations to limit barriers to uptake by the Vietnamese community by reinstating lifeguards at beach and lake sites, providing low-cost swim lessons and the development of translated material for services. In another study, Sandomierski and colleagues [85] reported that linking programs with a broader drowning prevention initiative may have created more opportunities for collaboration and consistency of messaging regarding child water safety.

#### 3.2.2. Engage the Target Group (c7)

Target group engagement included improving participation in an intervention by using existing networks and events [52,57,58,64,67,74,81]. For example, Franklin and colleagues [57] used schools to engage children, and Giresek [58] used pre-natal classes to engage new parents. Moran and Stanley [74] provided poolside education for parents while their children undertook swimming lessons. Mass media were used to disseminate information, such as updated boating safety regulations, to the general community in the study by Bugeja and colleagues [52]. 

##### Intervene with the Target Group to Enhance Uptake and Adherence (c7_02)

In the study by Petrass and colleagues [82], the community provided feedback on the timing of interventions, whilst Beattie, Shaw and Larson [50] received feedback on their intervention location, and Yusef and colleagues [93] reviewed evaluation tools used by pediatricians to educate parents.

#### 3.2.3. Evaluative and Iterative Strategies (c1)

Several (*n* = 6) drowning prevention pilot study interventions were identified [50,67,74,84,90,92]. The ongoing examination and refinement of implementation strategies by the intervention teams were also common in four studies [70,80,82,93]. Koon and colleagues [64] piloted school-based intervention materials based on lifeguards’ expertise in delivering an intervention and used focus groups with high school children to refine the program content and delivery [64].

##### Conducting Local Needs Assessment (c1_04)

Several articles (*n* = 11) described the use of local needs assessments to understand the target group. This included the use of focus groups (*n* = 6) [64,73,83,84,86,89], identification of barriers by experts (*n* = 2) via existing knowledge [75,92], review of the literature (*n* = 2) [77,81] and drowning trend data [95]. Morrongiello and colleagues [77] described using a review of the literature, local drowning trends and the National Water Safety Framework to inform the content for an awareness-raising program for parents on supervision of children around the water. Intervention locations were also purposively selected by need based on the formative findings [77]. In other articles, Mitchell and Haddrill [73] and Quan and colleagues [83] conducted focus groups with the local Chinese and Vietnamese–American communities, respectively. Savage and Franklin [86] conducted focus groups with culturally and linguistically diverse people regarding barriers to participating in water safety programs.

#### 3.2.4. Adapting and Tailoring the Context (c3)

Examples of ways in which interventions were adapted to suit local conditions included adaption to the availability of swimming facilities [80] and the swim ability of the target group [48,80,81]. The needs of the target group [79] and providers [64] and the accessibility and relevance of the interventions for specific communities [73,90] were considered. For example, Olaisen and colleagues [79] offered a variety of enrolment options for swimming lessons, providing between one and three swimming lessons. Petrass and Blitvich [81] tested perceived swim ability against actual swim ability in the first four lessons of their water-based intervention and then introduced specific skills relevant to the participants.

#### 3.2.5. Train and Educate Stakeholders (c5)

Stakeholder training and education were included in initial drowning prevention sessions (*conduct educational meetings (c5_01)*) to ensure key community leaders [66] and staff [69,79,82,91] understood the intervention before it was delivered to the wider community, as well as ongoing implementation meetings (*create a learning collaborative (c5_04)*) to ensure providers were learning from each other [93]. Other strategies included the development and distribution of resources for providers to support implementation [67,82,93] and the use of [45,79] and recommendations for [73,75] dynamic training delivery methods. For example, Love-Smith and colleagues [67] developed a presentation script and talking points for presenters at educational sessions, while Petrass and colleagues [82] developed lesson plans and other resources for providers to ensure the consistency of content delivery and lesson progression. The strategy to *provide ongoing consultation (C5_08)* was not identified in any articles.

#### 3.2.6. Provide Interactive Assistance (c2) and Support Providers (c6)

Only one strategy, *facilitation (c2_02)*, was identified [45,48,75,80,86]. Examples of *facilitation* included interactive problem solving undertaken with the target group, their support network and/or providers. For example, Olivar [80] identified that the swim teacher’s role was to identify issues in swimming skills with teaching styles focused on learner-centered problem solving with the participants. Teachers were considered active participants in the learning process and prioritized creating a supportive environment and opportunities were offered for participants to complete tasks at their own readiness and competency levels. The strategies *providing supervision (c2_04)*, *remind providers (C6_04)* and *revising professional roles (c6_05)* were not identified in the articles.

#### 3.2.7. Financial Strategies (c8) and Change Infrastructure (c9)

The concepts addressing *financial strategies (c8)* and *change infrastructure (c9)* were the least-frequently identified in the articles, with neither concept including strategies identified more than five times (key strategies). 

*Fund and contract for the evidence-informed intervention (c8_05)* was identified in two articles. Franklin and colleagues [57] identified the Swim and Survive Program as being subsidized by the Australian Capital Territory (ACT), and the Water Safety in the Bush project was funded by community organizations [50].

Van Weerdenburg, Mitchell and Wallner’s study [91] into pool fence compliance with the state’s Swimming Pools Act of 1992 in Australia described *financial strategies (c8)* and *change infrastructure (c9)*. *Changes were made to liability laws and enforcement (c9_03)* by granting authority to councils to access properties to inspect pools. Recommendations were made to correct or make explicit inconsistencies between the Act and other regulations and related Australian standards (*change liability laws or enforcement (c8_02)* and introduce a provision within the Act for inspection fees to assist with the cost of managing compliance, record systems and inspections (*place interventions on a fee-for-service list (c8_07)*). 

## 4. Discussion

This review sought to identify, describe and categorize the drowning prevention implementation strategies used in HIC settings. The findings were mapped to ERIC implementation concepts and strategies [7,11,41] to capture the breadth of implementation of drowning prevention interventions in HICs published in the peer-reviewed literature. It included 49 articles about drowning prevention interventions in HICs published between 2002 and 2022. The review found that articles were mostly from Australia and the USA, varying by sample and intervention level, with most interventions delivered at the group (e.g., school classroom, expectant parents) and population levels. Interventions most frequently used behavioral [42] strategies or a combination of behavioral and socio-ecological strategies [43]. Interventions covered the drowning prevention activities identified by the International Life Saving Federation [27], including environmental modifications, promoting swimming and lifesaving skills, cardiopulmonary resuscitation skills, surveillance and supervision. Evaluation designs were mostly quantitative, with several mixed-method and observational studies also included. All nine ERIC concepts and forty-two ERIC strategies were identified within the intervention studies. Fifteen strategies across six concepts were identified five times or more (key strategies). 

### 4.1. Understanding the Use of Implementation Strategies in Drowning Prevention Interventions

Three concepts were consistently identified: *developing partner relationships*, *engage target group* and *iterative and evaluative strategies*. This indicates that “developing relationships, engaging with the target group and checking what works as programs progress” are at the forefront for researchers and practitioners when reporting on drowning prevention activities. These strategies are also core competencies for those working in public health, health promotion [42,96], injury prevention [97] and advocacy [98], indicating that a complementary public health and injury prevention lens is used to frame drowning prevention interventions in HICs. This aligns with the way drowning prevention is framed as a public health issue by the World Health Organization [30,31,32,33,99]. The development of skills related to relationship building, target-group engagement and advocacy are a priority for the public health workforce, as central capabilities highlighted in the Global Charter for Public Health [100] and the Council of Academic Public Health Institutions Australasia (CAPHIA) Master of Public Health competencies [101].

Our review highlights that the fundamental principles of planning and evaluating programs (i.e., the concepts of formative research and stakeholder and target group engagement) are clearly identifiable components of the peer-reviewed drowning prevention literature in HICs. However, while these concepts were consistently identified, there was not always sufficient detail [102] available to support those implementing future interventions to replicate or decide [103] to use similar strategies and limited examples of implementation strategies described in the context of recommendations for similar programs. For example, Stempski and colleagues [89] stated that partner organizations identified a representative (project champion) who shared survey learnings at bimonthly meeting to help foster change among others as a collaborative and iterative process of improvement highlighting success and barriers, but did not explain how project champions were identified or engaged. To build practitioner implementation capacity, understanding what was carried out may not always be enough [104]. The implementation strategies used need to be compared and/or assessed against current practice to improve uptake [7]. 

This review highlights a need for the processes related to the transformation and adaptation of implementation strategies used in practice to be better understood [104]. One way to do this may be through the use of causal loop diagrams and system modelling [103] in drowning prevention interventions to better explore the interactions influencing decision making [104] and allow for an exploration of factors, such as organizational and evaluation capacity [102], affecting implementation decision making [7]. Alternatively, exploration of the gaps and factors affecting the uptake of implementation strategies [105] with practitioners and researchers would also be beneficial.

*Developing partner relationships* was the most frequently identified ERIC concept; *developing academic partnerships* was the most cited ERIC strategy. It is posited that the requirement for HREC approval when publishing in peer-reviewed journals may mean the impact of academic partnerships in the implementation of drowning prevention interventions in HICs is over-estimated in the literature. Research has found that practitioners often feel intimidated by the term “ethics”, equating oversight processes to research, and feel that the process of gaining ethics approval has limited benefit to service delivery [106,107]. Other identified barriers to the use of institutional ethical approval include organizational capacity, competing priorities and access [106]. These perceptions have the consequent effect of limiting the integration of ethical oversight into policy and practice [108] and further, reducing the likelihood of practitioners and policy makers publishing in the peer-reviewed literature, as ethical oversight is a requirement for most journals [109]. Being explicit about the ethical foundations of public health interventions are important to ensure they are informed by evidence, do what was intended, avoid iatrogenic effects and follow agreed guidelines and principles related to the ethical conduct of human research [107]. 

Drowning prevention interventions are designed and delivered by water safety practitioners, who have varying capacity and skills [97,110] in designing, implementing and evaluating programs. For example, practitioners in the water safety space are often lifeguards, fishers and swim instructors [110] working in complex settings and impacted by system factors that affect the intervention, provider and community [111,112], with variations in organizational capacity, staff skills and technical components [113]. Thus, consistent with findings in community health promotion more broadly [106], it is likely that drowning prevention practitioners may feel ethical approval has a limited benefit to service delivery. Enhancing knowledge of ethical practice and streamlining access to ethical oversight by making research–practice partnerships more common may facilitate greater participation in formal ethical oversight processes and greater contribution to the peer-reviewed literature by a broader range of practitioners.

Community and academic partnerships have the greatest potential to improve the successful implementation of evidence-informed practice [114]. Community and academic partnerships ensure that decision-making processes and subsequent interventions are feasible and sustainable [115,116] by utilizing a shared vision and impact benchmarking. To ensure that researchers consider interesting, important research questions and use effective methodology [114], practitioners develop evaluation practices and skills [117] and knowledge translation [106] within the industry occurs in a timely manner, there is a need to further develop strategies that truly enhance research–practice partnerships.

Overall, the review identified a lack of consistent language used to describe implementation of the interventions. For example, in the five articles [45,48,75,80,86] where examples of *facilitation (c2_02)* (a process of interactive problem solving and support that occurs in the context of a recognised need for improvement and a supportive interpersonal relationship [11]) were identified, the terms facilitation or problem solving were not used. Instead, examples described how the participants “engaged in tasks that targeted their underlying deficit” [45], described how “factors such as personal instructor qualities, program structure and support and day-to-day interactions with students were important” [48] and iterated that “time was allowed for the pool-side parents to seek advice from the instructor” [75]. Similar issues of inconsistent terminology have been identified by other authors when reviewing the obesity literature [118] and implementation guidelines more broadly [119]. More consistent use of implementation terminology in the drowning prevention literature would be useful to ensure that the implementation strategies are easier to identify and better understood. This could be achieved with the development and use of a framework guide for the implementation of drowning prevention interventions for use by practitioners, researchers, funding bodies and decision makers.

### 4.2. Gaps in the Use and Reporting of Implementation Strategies

Approximately 40% of the ERIC implementation strategies (*n* = 30) were not identified in included articles. This result is consistent with observational and qualitative research into the use of ERIC strategies in general practice across the USA [120]. We speculate that some non-identified ERIC strategies are likely to be undertaken but are limited in detail or not formally captured and reported in the peer-reviewed literature. This is highlighted by the limited reporting of *informing local opinion leaders (c4_08)* (*n* = 2) and *identifying and preparing champions (c4_06)* (*n* = 1) and no cases of *identifying early adopters (c4_07)*, despite community groups identified as partners and collaborators in multiple articles [48,49,73,83,84] and included in pre-delivery training to provider and community members [66]. The limited detail included in the literature, whereby the definitions for the ERIC strategies have not been met (e.g., identify and prepare champions), has been noted by other studies discussing the use of implementation strategies in community settings [7] and may also be the case for other strategies such as *remind providers (c6_04)* and *provide ongoing consultation (C5_08).*

The concepts of *financial strategies* and *change infrastructure* did not include any key strategies and were largely absent from the reviewed literature. Drowning prevention interventions tend to focus on education, the physical environment or community and social context, with *financial strategies (c8)* mainly utilized by interventions undertaken by local councils (in the case of pool fencing requirements) [91] and involving service delivery (i.e., access to pool facilities [83] or subsidized swim lesson participation [57]). In the case of *financial strategies*, it may be that the use of public health [26,31,99] rather than implementation science [7] to frame drowning prevention means that the economic impacts on behaviour change [121] and financial indicators of organizational capacity [122] have been somewhat overlooked [121]. 

### 4.3. What Was Learnt?

In general, the current use and reporting of implementation strategies in the published literature highlights that various implementation strategies are likely over-reported (reported in the literature at a higher proportion than they are used), under-reported (used more often than they are reported in the literature) and in some cases, overlooked (neither likely used nor reported in the literature). The consequence of over-reporting implementation strategies is a false sense of what is occurring in the field whilst under-reporting means interventions are difficult to replicate. Those publishing drowning prevention interventions in the peer-reviewed literature could refocus efforts towards intervention implementation. Further exploration of the use and adaptation of existing resources and systems (e.g., databases, provider training) to support providers to deliver interventions could go some way to support and strengthen the implementation of drowning prevention interventions. 

## 5. Strengths and Limitations 

Whilst the call for methodologically sound reviews of implementation in public health interventions has been made [7,119], this review is the first to report on ERIC implementation concepts and strategies using the HIC drowning prevention literature as a case study. Strengths include searching eight databases, a purposefully broad scope (i.e., did not include “interventions” in the search terms), following procedures for previously published systematic reviews [22,23,24] and the use of a public health-specific quality-appraisal tool (Meta QAT) [39]. Several limitations included the restriction to English language and the exclusion of the grey literature. The grey literature may have yielded a wider range of interventions; however, technical reports [123], annual reports [124] and websites [125] were more likely to describe interventions for funders and the general community and were deemed unlikely to describe implementation strategies. An over-representation of from Australia and the USA may reflect that drowning prevention efforts have attracted funding and resources, which has allowed for a research–practice nexus to be established and afforded peer-reviewed publications. In contrast, there were few non-English articles, suggesting drowning prevention may be a lower priority for research funding in some countries and the opportunity to publish becomes limited. As with other reviews of public health intervention implementation [118], the lack of consistent terminology to describe implementation strategies in the drowning prevention literature may mean some articles were missed. Despite these limitations, this study begins a discussion of the use of implementation science in drowning prevention interventions and adds to the small but growing evidence base on how drowning prevention interventions are implemented. 

## 6. Conclusions

The findings of this systematic review serve as a starting point for further exploration of the implementation strategies used in drowning prevention interventions. The review highlights the need for more detailed, accurate reporting of the implementation of interventions to aid in the replication and refinement of evidence-informed interventions. The use and reporting of implementation strategies in published, peer-reviewed drowning prevention interventions in HICs is varied and lacks depth, making interventions difficult to replicate or making it difficult to know which implementation strategies add to the success of an intervention and why. The concepts of evaluative and iterative strategies and adapting to the context are relatively well-developed. However, there is a paucity of evidence on other concepts such as how providers and stakeholders are supported, trained and educated. 

Potential improvements that may support better capture of the implementation strategies include increased articles in the peer-reviewed literature that describe the process of program planning, implementation and evaluation and the use of consistent drowning prevention implementation language. Supporting practitioners to identify and apply implementation strategies in their day-to-day work can facilitate real-world enhancements in public health action for drowning prevention in HICs.

Future endeavors include an exploration of intervention implementation with drowning prevention practitioners and researchers, which will allow for the gaps identified in this review to be further understood. We anticipate that this will go some way to better describe the use of implementation strategies, which has theoretical, methodological and practical implications, thus strengthening the implementation of evidence-informed interventions in HICs. 

## Figures and Tables

**Figure 1 ijerph-21-00045-f001:**
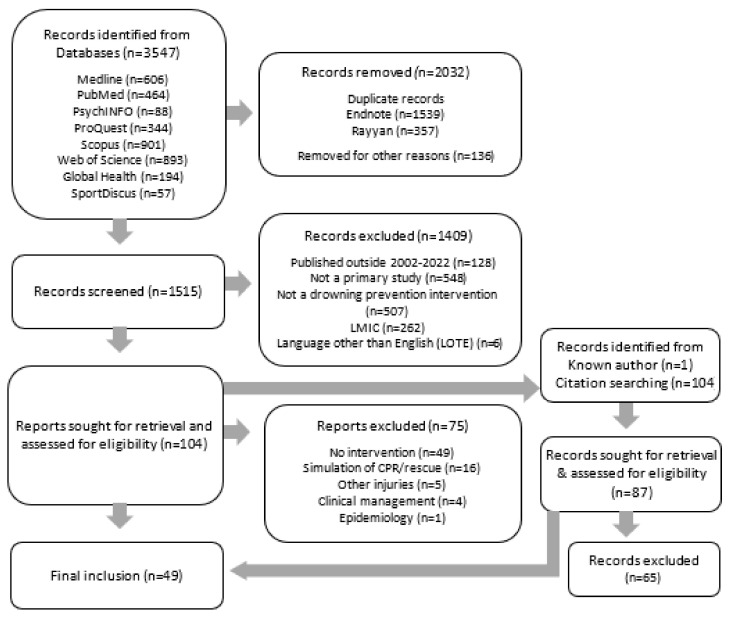
Flow diagram for review of implementation concepts and strategies identified in drowning prevention literature from 2002–2022.

**Table 1 ijerph-21-00045-t001:** Inclusion and exclusion criteria.

Inclusion Criteria	Exclusion Criteria
Primary studies	Clinical management
Conducted in HICs	Guideline and recommendation development
Drowning prevention intervention	Not relevant to the wider population (i.e., involves specific target group or location)
Written in English	Other injury focus
Published between 2002 and 2022	Epidemiological focus

**Table 2 ijerph-21-00045-t002:** ERIC strategy definitions.

Concepts and Strategies	Definition	Identifier
CONCEPT 1: Use evaluative and iterative strategies		c1
Assess for readiness and identify barriers and facilitators	Assess various aspects of an organization to determine its degree of readiness to implement, barriers that may impede implementation and strengths that can be used in the implementation effort.	c1_01
Audit and provide feedback	Collect and summarize intervention performance data over a specified time period and give it to providers and administrators to monitor, evaluate and modify provider behaviour.	c1_02
Conduct cyclical small tests of change	Implement changes in a cyclical fashion using small tests of change before taking changes system-wide. Tests of change benefit from systematic measurement, and results of the tests of change are studied for insights on how to do better. This process continues serially over time, and refinement is added with each cycle.	c1_03
Conduct local needs assessment	Collect and analyse data related to the need for the intervention (with the target group or their support network).	c1_04
Develop a formal implementation blueprint	Develop a formal implementation blueprint that includes all goals and strategies. The blueprint should include the following: (1) aim/purpose of the implementation; (2) scope of the change (e.g., what organizational units are affected); (3) timeframe and milestones; and (4) appropriate performance/progress measures. Use and update this plan to guide the implementation effort.	c1_05
Develop and implement tools for quality-monitoring	Develop, test and introduce into quality-monitoring systems the right input—the appropriate language, protocols, algorithms, standards and measures (of processes, patient/consumer outcomes and implementation outcomes) that are often specific to the intervention being implemented.	c1_06
Develop and organize quality-monitoring systems	Develop and organize systems and procedures that monitor intervention processes and/or outcomes for the purpose of quality assurance and improvement.	c1_07
Obtain and use target group and support network feedback	Develop strategies to increase target group and support network feedback on the implementation effort.	c1_08
Purposely re-examine the implementation	Monitor progress and adjust intervention practices and implementation strategies to continuously improve the quality of the intervention.	c1_09
Stage implementation scale-up	Phase implementation efforts by starting with small pilots or demonstration projects and gradually move to a system-wide rollout.	c1_10
CONCEPT 2: Provide interactive assistance		c2
*Centralise technical assistance*	*Develop and use a centralised system to deliver technical assistance focused on implementation issues.*	*c2_01*
Facilitation	A process of interactive problem solving and support that occurs in the context of a recognized need for improvement and a supportive interpersonal relationship.	c2_02
Provide technical assistance	Develop and use a system to deliver technical assistance focused on implementation issues using local personnel.	c2_03
Provide supervision	Provide providers with ongoing supervision focusing on the intervention. Provide training for provider supervisors who will supervise providers of the intervention.	c2_04
CONCEPT 3: Adapt and tailor context		c3
Promote adaptability	Identify the ways an intervention can be tailored to meet local needs and clarify which elements of the intervention must be maintained to preserve fidelity.	c3_01
Tailor strategies	Tailor the implementation strategies to address barriers and leverage facilitators that were identified through earlier data collection.	c3_02
*Use data experts*	*Involve, hire and/or consult experts to inform management on the use of data generated by implementation efforts.*	*c3_03*
*Use data warehouse techniques*	*Integrate intervention records across facilities and organizations to facilitate implementation across systems.*	*c3_04*
CONCEPT 4: Develop partner relationships		c4
Build a coalition	Recruit and cultivate relationships with partners in the implementation effort with partners involved throughout the whole intervention.	c4_01
Capture and share local knowledge	Capture local knowledge from implementation sites on how implementers and providers made something work in their setting and then share it with other sites.	c4_02
Conduct local consensus discussions	Collect and analyse data related to the need for the innovation (carried out with stakeholders).	c4_03
Develop academic partnerships	Partner with a university or academic unit for the purposes of shared training and bringing research skills to an implementation project.	c4_04
Develop an implementation glossary	Develop and distribute a list of terms describing the intervention, implementation and stakeholders in the organizational change.	c4_05
Identify and prepare champions	Identify and prepare individuals who dedicate themselves to supporting, marketing and driving through an implementation, overcoming indifference or resistance that the intervention may provoke in an organization or intended community.	c4_06
Identify early adopters	Identify early adopters at the local site to learn from their experiences with the practice intervention.	c4_07
Inform local opinion leaders	Inform providers identified by colleagues as opinion leaders or “educationally influential” about the intervention in the hopes that they will influence colleagues to adopt it.	c4_08
Involve executive boards	Involve existing governing structures (e.g., boards of directors, medical staff boards of governance) in the implementation effort, including the review of data on implementation processes.	c4_09
Model and simulate change	Model or simulate the change that will be implemented prior to implementation.	c4_10
Obtain formal commitments	Obtain written commitments from key partners that state what they will do to implement the intervention.	c4_11
Organise implementation team meetings	Develop and support teams of providers who are implementing the intervention and give them protected time to reflect on the implementation effort, share lessons learned and support one another’s learning.	c4_12
Promote network collaboration	Identify and build on existing high-quality working relationships and networks within and outside the organization, organizational units, teams, etc. to promote information sharing, collaborative problem solving and a shared vision/goal related to implementing the intervention.	c4_13
Recruit, designate and train for leadership	Recruit, designate and train leaders for the change effort.	c4_14
Use advisory boards and workgroups	Create and engage a formal group of multiple kinds of stakeholders to provide input and advice on implementation efforts and to elicit recommendations for improvements.	c4_15
Use an implementation advisor	Seek guidance from experts in implementation.	c4_16
Visit other sites	Visit sites where a similar implementation effort has been considered successful.	c4_17
CONCEPT 5: Train and educate stakeholders		c5
Conduct educational meetings	Hold meetings targeted toward different stakeholder groups (e.g., providers, administrators, other organizational stakeholders, target group and support network stakeholders) to teach them about the intervention.	c5_01
Conduct educational outreach visits	Have a trained person meet with providers in their provider settings to educate providers about the intervention with the intent of changing the provider’s practice.	c5_02
Conduct ongoing training	Plan for and conduct training in the intervention or implementation in an ongoing way.	c5_03
Create a learning collaborative	Facilitate the formation of groups of providers or provider organizations and foster a collaborative learning environment to improve implementation of the intervention.	c5_04
Develop educational materials	Develop and format manuals, toolkits and other supporting materials in ways that make it easier for stakeholders to learn about the intervention and for providers to learn how to deliver the intervention.	c5_05
Distribute educational materials	Distribute educational materials (including guidelines, manuals and toolkits) in person, by mail and/or electronically.	c5_06
Make training dynamic	Vary the information delivery methods to cater to different learning styles and work contexts, and shape the training in the intervention to be interactive.	c5_07
Provide ongoing consultation	Provide ongoing consultation with one or more experts in the strategies used to support implementing the intervention.	c5_08
Shadow other experts	Provide ways for key individuals to directly observe experienced providers engage with or use the targeted intervention.	c5_09
Use train-the-trainer strategies	Train designated providers or organizations to train others in the intervention.	c5_10
Work with educational institutions	Encourage educational institutions to train providers in the intervention or implementation.	c5_11
CONCEPT 6: Support providers		c6
Create new provider teams	Change who serves on the provider team, adding different disciplines and different skills to make it more likely that the intervention is delivered (or is more successfully delivered).	c6_01
Develop resource-sharing agreements	Develop partnerships with organizations that have resources needed to implement the intervention. Includes cases whereby existing resources were used but no sharing agreement was mentioned.	c6_02
Facilitate relay of program data to providers	Provide as close to real-time data as possible about key measures of process/outcomes using integrated modes/channels of communication in a way that promotes use of the targeted intervention.	c6_03
Remind providers	Develop reminder systems designed to help clinicians to recall information and/or prompt them to use the clinical innovation.	c6_04
Revise professional roles	Shift and revise roles among professionals who provide care, and redesign job characteristics.	c6_05
CONCEPT 7: Engage target group		c7
Increase demand	Attempt to influence the market for the intervention to increase competition intensity and to increase the maturity of the market for the intervention.	c7_01
Intervene with target group to enhance uptake and adherence	Develop strategies with the target group and/or their support network to encourage and problem solve around adherence.	c7_02
Involve target group and support network	Engage or include the target group and/or their support network in the implementation effort.	c7_03
Prepare target group to actively participate	Prepare the target group to be active in their involvement, to ask questions and specifically, to inquire about guidelines and the evidence behind the intervention.	c7_04
Use mass media	Use media to reach large numbers of people to spread the word about the intervention.	c7_05
CONCEPT 8: Financial strategies		c8
Access new funding	Access new or existing money to facilitate the implementation.	c8_01
Alter provider incentives/allowance structures	Work to incentivize the adoption and implementation of the intervention.	c8_02
Alter fees for target group	Create fee structures where the target group or their support network pay less for the intervention (e.g., community pool access).	c8_03
Develop disincentives	Provide financial disincentives for failure to implement or use the intervention.	c8_04
Fund and contract for the evidence-informed intervention	Governments and other payers of services issue requests for proposals to deliver the intervention, use contracting processes to motivate providers to deliver the intervention and develop new funding formulas that make it more likely that providers will deliver the intervention.	c8_05
*Make billing easier*	*Make it easier to bill for the intervention.*	*c8_06*
Place interventions on a fee-for-service list/formularies	Work to place the intervention on lists of actions for which providers can be reimbursed (e.g., pool fence compliance checks charged to pool owners).	c8_07
*Use capitated payments*	*Pay providers a set amount per target group member for delivering the intervention.*	*c8_08*
*Use other payment schemes*	*Introduce payment approaches (in a catch-all category).*	*c8_09*
CONCEPT 9: Change infrastructure		c9
*Change accreditation or membership requirements*	*Strive to alter accreditation standards so that they require or encourage use of the intervention. Work to alter membership organization requirements so that those who want to affiliate with the organization are encouraged or required to use the intervention.*	*c9_01*
Change liability laws or enforcement	Participate in liability reform efforts that make providers more willing to deliver the intervention.	c9_02
Change physical structure and equipment	Evaluate current configurations and adapt, as needed, the physical structure and/or equipment (e.g., changing the layout of a room, adding equipment) to best accommodate the targeted intervention.	c9_03
Change record systems	Change records systems to allow better assessment of implementation outcomes.	c9_04
Change service sites	Change the location of service sites to increase access.	c9_05
*Create or change credentialing and/or licensure standards*	*Create an organization that certifies providers in the intervention or encourage an existing organization to do so. Change governmental professional certification or licensure requirements to include delivering the intervention. Work to alter continuing education requirements to shape professional practice toward the intervention.*	*c9_06*
Mandate change	Have leadership declare the priority of the intervention and their determination to have it implemented.	c9_07
Start a dissemination organization	Identify or start a separate organization that is responsible for disseminating the intervention. It could be a for-profit or non-profit organization.	c9_08

*Italics* indicate strategies of a technical/clinical nature, not expected to be seen in the literature.

**Table 3 ijerph-21-00045-t003:** General study characteristics, intervention, evaluation and implementation (*n* = 49).

Article Details	Sample	Intervention	Evaluation	Implementation
Author: Alaniz, Rosenberg, Beard and Rosario Year: 2017 Aim: To examine the feasibility and effectiveness of an aquatic therapy program Location: USA	Sample: Children (aged 3–7 years) with mild to severe autism spectrum disorder (*n* = 7) Male (*n* = 6) Female (*n* = 1) Recruitment: Purposive sampling Response rate: N/R	Intervention Level: Individual Type: Behavioural Activity: Education Water safety lessons: 1 h/week × 8 1 h/week × 16 1 h/week × 24 Duration: Approximately 6 months Theory: N/R Formative research: No	Design: Observational In situ pre- and post-observation Measures: Swim skills Social skills Ethical approval: Yes Findings: Skills Improvement of overall water safety skills. No difference in social skills, baseline to 8 h intervention.	Level: Provider Concepts and Strategies: c2. Provide interactive assistance Facilitation c3. Adapt and tailor context Tailor strategies c5. Train and educate stakeholders Use train-the-trainer strategies
Author: Araiza-Alba, Keane, Matthews, Simpson, Strugnell, Sun Chen and Kaufman Year: 2021 Aim: To investigate the effectiveness and engagement of virtual reality (VR) video tools to enhance learning about coastal water safety skills and hazard identification Location: Australia	Sample: School-aged children (10–12 years) (*n* = 182) Male (*n* = 67) Female (*n* = 48) Recruitment: N/R Response rate: 62.7%	Intervention Level: Group Type: Behavioural Activity: Education VR technology for learning Duration: 10 weeks (Test 1 data collection to Test 4 data collection) Theory: Cognitive Theory of Multimedia Learning and Interest Formative research: No	Design: Mixed methods; Survey and focus groups Measures: Knowledge of water safety behaviours Hazard identification Activity interest and enjoyment Ethical approval: Yes Findings: Interest and enjoyment Significant differences between VR and TV and VR and poster. 15.2% of VR group participants reported high levels of presence. Knowledge No significant difference across the three conditions (VR, TV and poster). Learning scores increased for 94.7% of participants.	Level: Organization Concepts and Strategies: c4. Develop partner relationships Develop academic partnerships
Author: Bauer, Giles, Marianayagam and Toth Year: 2020 Aim: Outline the adoption of the Kids Don’t Float program into a more family-centred approach Location: USA	Sample: Parents and children residing in Alaska (*n* = 12,846) Recruitment: N/R Response rate: N/R	Intervention Level: Population Strategy Type: Mixed Activity: Environmental Loaner board program provided free access to LJ. Education Cold water and boating safety lessons, migrant education program, school education and in-water family events Duration: Program has been running for 20 years Theory: N/R Formative research: No	Design: Not identified Ethical approval: N/R Findings: Drowning rates Drowning incidents in Alaska have decreased since the adoption of a family-centred approach, 2019 (*n* = 11) compared to 1999 (*n* = 26).	Level: Target group Concepts and Strategies: c7. Engage target group Intervene with target group to enhance uptake and adherence
Author: Beale Year: 2012 Aim: To describe a collaboration designed to provide high-quality health and physical activity programs for youth in underserved communities, including Project Guard: Make A Splash END (End Needless Drowning) Location: USA	Sample: Middle and high school students (*n* = 300) in a high-needs public school district (mostly low-SES and CALD communities) in New York State Recruitment: N/R Response rate: N/A	Intervention Level: Group Type: Mixed Activity: School-based learning experience involving: Provision of introductory materials Student goal setting for lifeguard, CPR and first aid training Provision of equipment and resources Lifeguard training undertaken Snack and transportation home Employment opportunities after course completion Duration: 3 years Theory: Humanistic and social curricular model—teaching for personal and social responsibility Educational learning standards for physical education and career and technical rducation Formative research: No	Design: Process evaluation Measures: Swimming certificate achievements First aid and CPR certification Lifeguarding and water safety instructor certification Attitudes Article mentions the use of: Program-related documents, Field notes from observations, Focus group interviews, Formal and informal interviews of individual key stakeholders Ethical approval: N/R Findings: Skills 100% of participants received one or more American Red Cross instructional swimming certificates Relational changes in interaction with participants	Level: Organization, target group Concepts and Strategies: c1. Use evaluative and iterative strategies Assess for readiness and identify barriers and facilitators c2. Provide interactive assistance Facilitation c3. Adapt and tailor context Tailor strategies c4. Develop partner relationships Build a coalition Promote network collaboration
Author: Beale-Tawfeeq, Anderson and Ramos Year: 2018 Aim: Reflective analysis: Water safety-based education and water safety messaging by a national aquatic stakeholder New collaborations with diverse partners The coordination of a national effort to raise public awareness. Location: USA	Sample: Stakeholders (*n* = 10) working with CALD communities Recruitment: N/R Response rate: N/R	Intervention Level: Group Type: Socio-ecological Activity: National Leaders’ Summit on Diversity in Aquatics held to develop action plans and toolkits with stakeholder members to aid in multisectoral collaborations by: Understanding of history affecting CALD drowning Development of strategies to encourage involvement of CALD communities Enhance partnerships through active engagement with CALD communities Duration: 1 day with ongoing relationships formed Theory: A social justice lens to inform programming and collaborative partnerships Formative research: No	Design: Process evaluation Measures: Number and type of stakeholders involved in the National Leaders Summit Ethical approval: N/R Findings: Stakeholder organizations developed water safety objectives, examples given from one organization Stakeholder involvement in International Water Safety Day activities Some recommendations for working with CALD communities are discussed Long term outcomes from the intervention are not clearly discussed	Level: Organization Concepts and Strategies: c4. Develop partner relationships Promote network collaboration c5. Train and educate stakeholders Conduct educational meeting
Author: Beattie, Shaw and Lawson Year: 2008 Aim: To describe the implementation and evaluation of a water safety training program: “Water Safety in the Bush” Location: Australia	Sample: Adults (*n* = 219) and children (*n* = 668) residing in rural and remote communities (*n* = 887) Recruitment: Advertised and recruited through networks Response rate: Community organizations 48.3% (*n* = 14); (*n* = 29) expression of interest forms Parental satisfaction survey 10% (*n* = 82)	Intervention Level: Group Type: Mixed Activity: Educational Swimming and water safety instruction program Training community members as AUSTSWIM teachers Environmental Water safety signage Provision of resources within community Duration: Approx. 1–2 years Theory: Behavioural–ecological model Formative research: Yes, community consultation to develop strategies	Design: Mixed methods Pre- and post-test Analysis of instructor notes Measures: Parental satisfaction Swimming skills Resource dissemination First aid training certification AUSTSWIM teaching accreditation Ethical approval: N/R Findings: Skills and behaviour Feedback from instructors and parents indicated there was progress in swimming skills among the participants at all sites. Adults and older children received first aid training (*n* = 47) and community members (*n* = 38) became AUSTSWIM-accredited instructors. Environment All sites installed water safety signage at local water hazards.	Level: Organization, provider, target group Concepts and Strategies: c1. Use evaluative and iterative strategies Stage implementation scale-up c3. Adapt and tailor context Promote adaptability Tailor strategies c4. Develop partner relationships Inform local opinion leaders c6. Support providers Create new provider teams Develop resource sharing agreements c8. Financial strategies Fund and contract for the evidence-informed intervention c9. Change infrastructure Change record systems Change service sites Recommendations: c1. Use evaluative and iterative strategies Assess for readiness and identify barriers and facilitators c4. Develop partner relationships Recruit, designate and train for leadership
Author: Brander, Williamson, Dunn, Hatfield, Sherker and Hayden Year: 2022 Aim: Assess the effectiveness of the Science of the Surf (SOS) presentation in improving rip current knowledge, identification skills and decision-making regarding rip currents and beach flags Location: Australia	Sample: Community members aged 14+ years (*n* = 601) Male: 48.6% Female: 51.4% Recruitment: Advertised online, community newspaper Response rate: Pre- and post-surveys: (*n* = 256), 63.7% Follow-up (those who provided contact details): (*n* = 121), 52.6%	Intervention Level: Group Type: Behavioural Activity: Education 50 min presentation Duration: Approx. 1 month 11 January–8 February 2009 Theory: N/R Formative research: No	Design: Quantitative Pre- and post-test (three wave) Measures: Beach safety, hazards and rip current knowledge Ability to identify a safe swimming location Ability to identify a rip current Ethical approval: Yes Findings: Knowledge Rip current identification At post and follow-up, no participants reported “don’t know” when asked what a rip current looks like Significant improvements in recalling correct rip current identifiers Behavioural intentions Post-intervention, a significant increase in participants indicating they would not swim at a beach if it was unpatrolled	Level: Organization Concepts and Strategies: c4. Develop partner relationships Develop academic partnerships
Author: Bugeja, Cassell, Brodie and Walter Year: 2014 Aim: To investigate whether the 2005 PFD wearing regulations reduced drowning deaths among recreational boaters Location: Australia	Sample: Recreational boating cases, deceased by drowning in Victoria Pre (*n* = 59); Post (*n* = 16) Male (*n* = 58) Female (*n* = 1) Recruitment: coronial data Response rate: N/A	Intervention Level: Population Type: Socio-ecological Activity: Regulatory Regulations required all occupants of recreational vessels to wear a specified type of PFD during designated times and scenarios. Duration: Approximately 11 years Theory: N/R Formative research: No	Design: Quantitative Retrospective Measures: Drowning deaths among recreational boaters PFD wearing status Ethical approval: Yes Findings: Drowning deaths Significant decrease in drowning deaths among all recreational boaters in Victoria. 59 pre-intervention compared to 16 post-intervention.	Level: Organization Concepts and Strategies: c4. Develop partner relationships Develop academic partnerships Promote network collaboration c5. Train and education Conduct educational meetings c7. Engage target group Use mass media
Author: Cassell and Newstead Year: 2015 Aim: To estimate the effect of the 2005 PFD wearing regulations on PFD use by occupants of small power recreational vessels. Location: Australia	Sample: Users of boat ramps in metropolitan and regional remote Victoria, aged 0–65+ years Occupants of small vessels. Pre (*n* = 1196), 44.6% Post (*n* = 1062), 45.2% Occupants of large vessels. Pre: (*n* = 1486), 55.4% Post: (*n* = 1285), 54.8% Recruitment: Observational Response rate: N/R	Intervention Level: Population Type: Socio-ecological Activity: Regulatory Regulations required all occupants of recreational vessels to wear a specified type of PFD during designated times and scenarios Duration: Approx. 2 yrs. January 2005–March 2007 Theory: N/R Formative research: No	Design: Observational Pre and post Measures: PFD use and type Vessel type and length Boater age and gender Number of people onboard Purpose and direction of trip Ethical approval: Yes Findings: Behaviour PFD use increased among occupants in: Small vessel 22% to 63% Large vessel 12% to 13%	Level: Organization, target group Concepts and Strategies: c1. Use evaluative and iterative strategies Develop and implement tools for quality monitoring c4. Develop partner relationships Develop academic partnerships c7. Engage target group Intervene with target group to enhance uptake and adherence Use mass media
Author: Casten, Crawford, Jancey, Della Bona, French, Nimmo and Leavy Year: 2022 Aim: To evaluate the proximal impact of two Keep Watch television commercials (TVC) on awareness, comprehension, acceptance and intention Location: Australia	Sample: Western Australian residents who care for a child <5 yrs. (*n* = 257) Age: 18–55+ yrs. Male (*n* = 9), 3.6%, Female (*n* = 238), 96.4% Recruitment: Dissemination via social media, networks and community events Response rate: N/R	Intervention Level: Population Type: Behavioural Activity: Education 12-week mass media campaign 30 s TVC 15 s TVC Duration: 23 weeks November 2017–April 2018 Theory: N/R Formative research: No	Design: Quantitative Post-test only Measures: Awareness Comprehension Acceptance Behavioural intention Campaign execution Ethical approval: Yes Findings Campaign awareness 32.3% were aware of the TVCs Comprehension 83% of respondents who recognised TVC1 and 82.1% of respondents who recognised TVC2 comprehended the message Acceptance Respondents reported over 90% acceptance of all four messages of TVC1, while two messages of TVC2 reported 84.4% acceptance from respondents Behavioural Intentions 50% of respondents who comprehended the TVC messages intended to take actions	Level: Organization, target group Concepts and Strategies: c1. Use evaluative and iterative strategies Conduct cyclical small tests of change Develop and implement tools for quality monitoring c4. Develop partner relationships Develop academic partnerships c7. Engage target group Use mass media
Author: Chung, Quan, Bennett, Kernic and Ebel Year: 2014 Aim: To assess LJ use among Washington State boaters and examine the relationship between LJ use and boating laws Location: USA	Sample: Washington boaters (*n* = 5157) Male 63.8% Female 36.2% Recruitment: N/A Response rate: N/R	Intervention Level: Population Type: Mixed Activity: Regulatory Requirement to wear LJs for people being towed by boats, occupants of personal watercraft and children aged 0–12 yrs old on boats. Duration: 2 months August–September 2010 Theory: N/R Formative research: No	Design: Observational In situ observation Measures: Demographics Type of boat LJ use Ethical approval: Yes Findings: Behaviour 30.7% of all observed occupants wore LJs LJ use was highest among individuals who were required to by law PWC occupants 96.8%; individuals towed by boats 95.3%; children: 81.7%	Level: System, organization, target group Concepts and Strategies: c1. Use evaluative and iterative strategies Develop and implement tools for quality monitoring Stage implementation scale-up c7. Engage target group Intervene with target group to enhance uptake and adherence Recommendations: c9. Change infrastructure Change liability laws or enforcement
Author: Evans and Sleap Year: 2013 Aim: To present key evaluation findings from the “Swim for Health” program Location: United Kingdom	Sample: Two Local Authority Areas Targets groups: Employed full-time Families with preschool-aged children (through local children’s centres (*n* = 10) Aged 50+ yrs With special health requirements Recruitment: Based on availability of aquatic sessions in the area Participants were primarily either young parents (group 2) or older women over the age of 50 years (groups 3 and 4). Response rate: N/R	Intervention Level: Group Type: Mixed Activity: Target group 1 Improved signposting of existing aquatic activity Increased uptake of corporate leisure membership opportunities Organise specific water-based sessions where appropriate Target group 2 Involve 1% of parents in aquatic activity Use of swim sessions/aquatic activity at or near capacity Additional activity to meet additional demands Target group 3 Offer appropriate aquatic activity Ensure that sessions at or near capacity Target group 4 Ensure inclusion of aquatic activity on exercise referral schemes Ensure that use of aquatic activity offered through referral schemes at or near capacity Duration: 3 years (2005–2008) Theory: N/R Formative research: No	Design: Mixed methods Record keeping and participation records Interviews with stakeholders Commissioned by Amateur Swimming Association Theory-driven scientific program evaluation completed by an evaluation officer Measures: Trends in program processes were noted, tracked and compared with the initial plans Chronological records of all developments within the program Participation records Service planning and provisions Triangulation with *n* = 20 semi-structured interviews with stakeholders, i.e., strategic and facility managers, service providers and exercise professionals. Ethical approval: N/R Findings: Due to the broad inclusion scope of target groups the program was a qualified success, although many participants were already participating in aquatic activity Misaligned rationale (to reduce health inequalities among its target groups), objectives (to increase aquatic activity participation), goals and performance indicators (no specific health message and no physiological or anthropometric measures to track health)	Level: Organization Concepts and Strategies: c4: Develop partner relationships Use advisory boards and workgroups c9. Change infrastructure Change record systems Recommendations: c1. Use evaluative and iterative strategies Assess for readiness and identify barriers and facilitators Conduct local needs assessment Purposely re-examine the implementation Stage implementation scale-up c3. Adapt and tailor context Promote adaptability c4. Develop partner relationships Conduct local consensus discussions
Author: Franklin, Peden, Hodges, Lloyd, Larsen, O’Connor and Scarr Year: 2015 Aim: To explore whether children can achieve the levels set by the Water Safety Education Competency Formative Research Framework and what factors impact the achievement of levels Location: Australia	Sample: Primary school children aged 5–12 years (*n* = 7726) in Canberra Male: 48.7% Female: 51.3% Recruitment: RLSS-ACT database Response rate: N/R	Intervention Level: Population Type: Behavioural Activity: Education Swim and Survive Program delivered by RRSS-ACT during school hours. Duration: 3 years Theory: N/R Formative research: No	Design: Quantitative Pre-test (Parent survey) Intervention data (swim skill records) Measures: Demographics School level Swim level achieved Swimming behaviours Medical conditions Experiences Ethical approval: N/R Findings: Skills As aged increased, average level achieved increased	Level: System, organization, target group Concepts and Strategies: c1. Use evaluative and iterative strategies Assess for readiness and identify barriers and facilitators c8. Financial strategies Fund and contract evidence-informed intervention
Author: Girasek Year: 2011 Aim: To evaluate whether a brief video could motivate pregnant pool owners to be trained in infant/child CPR Location: USA	Sample: Pregnant pool owners (*n* = 110) in Florida aged 18–47 years. Intervention (*n* = 76) Control (*n* = 34) Recruitment: Purposive sampling Response rate: 61% of eligible participants agreed to participate Follow-up 91.8%	Intervention Level: Group Type: Behavioural Activity: Educational Two videos (7 min and 9 min versions) included: Toddler drowning statistics Fear-based footage Discussion of advantages for pool fencing installation and infant CPR training Segment of a mother recounting personal experience of child drowning (9 min video) Duration: 3 yrs Nov 2005–Nov 2008 Theory: Health Belief Model Formative research: No	Design: Quantitative Pre- and post-test Quasi-experimental Measures: Attitudes and beliefs Behaviours Risk perception Ethical approval: Yes Findings: Beliefs Intervention group’s belief indicators (perceived susceptibility and severity) significantly increased from baseline to follow-up Behaviour Participants in the intervention group were significantly more likely to report current CPR training 6 months post-birth.	Level: Target group Concepts and Strategies: c7. Engage target group Increase demand
Author: Glassman, Castor, Karmakar, Blavos, Dagenhard, Domigan, Sweeney, Diehr and Kucharewski Year: 2018 Aim: To determine if a social marketing campaign guided by the Health Belief Model could improve parents’ (from the inner city) knowledge and perceptions concerning water safety Location: USA	Sample: Parents and legal guardians of children aged 7–16 yrs enrolled in swimming lessons (*n* = 65) in the Midwest Intervention (*n* = 39) Control (*n* = 26) Males (*n* = 5) Females (*n* = 58) Recruitment: N/R Response rate: 53% of participants who completed the pre-test completed the post-test	Intervention Level: Group Type: Behavioural Activity: Education Each week a new message/resource was disseminated to participants through brochures, email messages, postcards, window cling, Facebook messages and text messages. Each medium was linked to a concept of the Health Belief Model. Duration: 6 weeks Theory: Health Belief Model Formative research: Yes	Design: Quantitative Pre- and post-test Quasi-experimental Measures: Knowledge Perceptions Self-efficacy Behavioural intentions Ethical approval: Yes Findings: Knowledge Participants in treatment group improved knowledge and risk perception post-intervention	Level: Organization, target group Concepts and Strategies: c1 Use evaluative and iterative strategies Develop and implement tools for quality-monitoring c4. Develop partner relationships Promote network collaboration
Author: Hamilton, Keech, Wilcox-Pidgeon and Peden Year: 2022 Aim: To evaluate the impact of a video depicting a male discouraging his mates from mixing alcohol and aquatic activities Location: Australia	Sample: Males aged 18–34 years who consume alcohol and engage in aquatic activities (*n* = 97) Recruitment: Convenience Response rate: 48.5% response rate to follow-up (T3)	Intervention Level: Population Strategy Type: Mixed Activity: Education 30 s video designed as part of a media campaign “Don’t Let Your Mates Drink and Drown” Duration: N/R Theory: Theory of Planned Behaviour and Health Belief Model Formative research: Yes	Design: Quantitative Pre- and post-test (three wave) Measures: Behavioural intentions Attitudes Risk perceptions Subjective norms Ethical approval: Yes Findings Behavioural intentions and attitudes Influenced immediately post-intervention; however, this effect declined at follow-up Subjective norms Discouraging mates and self from consuming alcohol and swimming showed significant difference between test 1 and follow-up.	Level: Organization, target group Concepts and Strategies: c4. Develop partner relationships Develop academic partnerships c7. Engage target group Use mass media
Author: Hamilton, Peden, Keech and Hagger Year: 2018 Aim: To determine whether an infographic would have an impact on the beliefs and intentions of individuals driving through floodwater Location: Australia	Sample: Australian licensed drivers (*n* = 201) Age: 17–77 yrs Male (*n* = 41) Female (*n* = 160) Recruitment: Convenience sample Response rate: 83.8% response rate from T2 (post-intervention) and T3 (follow-up).	Intervention Level: Population Type: Behavioural Activity: Education 2 min and 11 s video infographic Duration: Approximately 4 months Theory: Mapped constructs on relevant behaviour change methods Models not explained. Formative research: No	Design: Quantitative Pre- and post-test (three wave) Measures: Behavioural intentions Attitudes Barrier self-efficacy Perceptions Ethical approval: Yes Findings: Male participants reported significantly higher intentions and attitudes towards driving through flood waters at all points Perceived susceptibility and perceived severity scores significantly increased post-intervention and were maintained at follow-up among female participants This effect was not reported among male participants as scores remained low at all time points	Level: Organization Concepts and Strategies: c4. Develop partner relationships Develop academic partnerships Promote network collaboration
Author: Hatfield, Williamson, Sherker, Brander and Hayen Year: 2012 Aim: To evaluate a campaign to improve beachgoer recognition of calm-looking rip currents Location: Australia	Sample: Beachgoers in Pacific Palms (intervention) and Mollymook (control) at three time points: Pre-intervention Intervention (*n* = 180) Control (*n* = 196) Post-intervention Intervention (*n* = 552) Control (*n* = 408) Follow-up Intervention (*n* = 222) Control (*n* = 161) Recruitment: Convenience sample Response rate: Varied from 54.9% (control site, follow-up); 83.7% (intervention site, pre-intervention)	Intervention Level: Population Type: Mixed Activity: Educational and environmental Distribution of “Don’t get sucked in by the rip” posters, postcards and brochures. Duration: Approximately 16 months April 2007–August 2008 Theory: Intervention based on cognitive theories of behaviour Formative research: Yes, interviews with beachgoers to inform campaign materials	Design: Quantitative Pre- and post-test (three wave) Quasi-experimental Measures: Behavioural intentions Behaviours Hazard identification Beliefs Campaign awareness Ethical approval: N/R Findings Awareness of campaign 28.8% of the post-intervention and 57.2% of follow-up respondents were aware of the “Don’t get sucked in by the rip” campaign Hazard Identification Intervention respondents were more likely to report correct swimming intentions and identification in relation to rip currents compared to the control respondents	Level: Organization, provider Concepts and Strategies: c7. Engage target group Use mass media
Author: Houser, Trimble, Brander, Brewster, Busek, Jones and Kuhn Year: 2017 Aim: To assess rip current knowledge to improve and enhance existing beach safety education material Location: USA	Sample: Adults (*n* = 1622) Texas (*n* = 368) North Carolina (*n* = 214) California (*n* = 184) Florida (*n* = 130) <50 respondents from other states. Male: 45% Female: 55% Recruitment: Convenience sample Response rate: 78% completed the survey	Intervention Level: Population Type: Behavioural Activity: Education, environmental “Break the Grip of the Rip” campaign involved dissemination through webpages, brochures, beach signage, videos, newspaper articles and public service announcements Duration: Campaign began in 2004, survey dissemination in May–August 2015. Theory: N/R Formative research: No	Design: Quantitative Post-test only Measures: Awareness and attitudes towards campaign Knowledge Hazard identification Perceptions Ethical approval: Yes Findings: Awareness 18% of respondents had heard of the campaign Knowledge 11% of respondents incorrectly selected the rip current as the safest place to swim when asked to identify on an image. 54% correctly identified the safest place to swim	Level: Organization Concepts and Strategies: c4. Develop partner relationships Build a coalition
Author: Koon, Brander, Alonzo and Peden Year: 2022 Aim: To explore and describe methodological aspects of the co-design process for the design and delivery of a beach safety education program Location: Australia	Sample: Year 7 students residing in the Lake Macquarie City Council (LMCC) region (*n* = 26) Students Male (*n* = 14) Female (*n* = 12) Expert survey (*n* = 11) Researchers (*n* = 3) LMCC lifeguards (*n* = 7) Program administrator (*n* = 1) LMCC Workshop Lifeguards (*n* = 8) Administrator (*n* = 1) Manager (*n* = 1) Recruitment: Purposive Response rate: N/R	Intervention Level: Group Type: Behavioural Activity: Education Co-designing a 45 min beach safety education presentation with lifeguards and students Duration: 2–3 months Theory: Theory of Planned Behaviour Formative research: Yes	Design: Mixed methods Survey, workshop and focus groups Measures: Content experts (opinions and recommendations) Student focus group (knowledge and attitudes of beach safety and behaviour) Ethical approval: Yes Findings: Attitude Student participants felt programs lacked reasoning and explanations behind safety advice. Student participants reported conflicting opinions regarding the use of storytelling and statistics Behaviour Student participants reported knowledge of risks and hazards when attending beaches; however, failed to report water safety behaviours. Peer influence was a common theme impacting decision making and behaviours	Level: Organization, provider, target Group Concepts and Strategies: c1. Use evaluative and iterative strategies Conduct local need assessment c3. Adapt and tailor context Tailor strategies c4. Develop partner relationships Build a coalition Capture and share local knowledge Develop academic partnerships c5. Train and educate stakeholders Work with educational institutions
Author: Koon, Bennett, Stempski and Blitvich Year: 2021 Aim: To identify safe water practices in the priority population and determine facilitators and barriers to behaviour change Location: USA	Sample: Low-income parents of preschool-aged children from culturally and linguistically diverse backgrounds (*n* = 90) in Washington Language spoken at home English (*n* = 40) Chinese (*n* = 35) Spanish (*n* = 8) Vietnamese (*n* = 6) Amharic (*n* = 1) Recruitment: Purposive Response rate: N/R	Intervention Level: Group Type: Behavioural Activity: Education One-hour education session held by community health educators and translators. One LJ per participating family was provided free of charge Duration: Approx. 2 yrs Theory: Fishbein’s integrated model of behavioural prediction Formative research: Yes	Design: Mixed methods Survey (pre- and post-test) Phone interview Measures: Skills and behaviour (swim ability and LJ use) Attitudes Behaviour (intent and actual) Knowledge Ethical approval: Yes Findings: Attitude 98% of participants reported feeling more confident keeping their children safe around water post-program Behaviour Post-program, 90% of participants reported intention to practice water safety behaviours including enrolment in swimming lessons and use of LJ. Only three participants recalled increased supervision. 53.7% of parents reported to have improved water safety behaviours post-program	Level: Providers, target group Concepts and Strategies: c6. Support providers Develop resource sharing agreements c7. Engage target group Intervene with target group to enhance uptake and adherence
Author: Lawson, Duzinski, Wheeler, Yuma-Guerrero, Johnson, Maxson and Schlechter Year: 2012 Aim: To evaluate a water safety curriculum in a low-income, minority-focused, urban youth summer camp Location: USA	Sample: Pre-K to third-grade students attending an urban youth summer camp (*n* = 166) Pre-K/kindergarten (Pre-K/K) (*n* = 33) First/second grade (*n* = 72) Third grade (*n* = 61) Recruitment: N/R Response rate: Consent received: 92.7% 3-week follow-up: 71.1%	Intervention Level: Group Type: Behavioural Activity: Education Danger Rangers water safety program The curriculum differed between age groups and ran for 4 hs/day for 1 week Duration: N/R Theory: N/R Formative research: No	Design: Quantitative Pre- and post-test (three waves) Measures: Parents and students surveyed Water safety knowledge Behaviour (post-test, observed by parents) Ethical approval: N/R Findings: Knowledge Each group scored higher post-test compared to pre-test At follow-up: Pre-K/K group did not score higher compared to pre-test. First/second and third grade scored higher	Level: Provider Concepts and Strategies: c5. Train and educate stakeholders Conduct educational meetings c7. Engage target group Involve target group and support network Recommendations: c3. Adapt and tailor context Promote adaptability Tailor strategies
Author: Love-Smith, Koon, Tabios and Bartell Year: 2022 Aim: To evaluate the efficacy of a brief educational drowning prevention program to increase self-reported water-safe behaviours, attitudes, self-efficacy and knowledge in parents and caregivers attending children’s swim lessons Location: USA	Sample: Parents and caregivers of children attending swim lessons (*n* = 443) in California Male 18% Female 82% Recruitment: N/R Response rate: Collected *n* = 172 pre-program surveys; *n* = 128 contacted for follow-up; *n* = 44 completed post-program survey	Intervention Level: Group Type: Behavioural Activity: Education Eyes Save Lives water safety program is a 12–15-min presentation Duration: 2 months July–August 2018 Theory: Social cognitive theory, Health Belief Model and Theory of Planned Behaviour Formative research: No	Design: Quantitative Pre- and post-test Measures: Attitudes Subjective norms Self-efficacy Self-reported behaviour Water safety knowledge Ethical approval: Yes Findings Attitudes Attitudes towards maintaining constant supervision of children in a pool did not alter Attitudes that support adults being within an arm’s reach of children within the water significantly increased Self-efficacy Reported self-confidence in responder to an emergency in the water increased Knowledge Composite knowledge scores increased between pre and post	Level: Provider, target group Concepts and Strategies: c1. Use evaluative and iterative strategies Obtain and use target group and support network feedback Stage implementation scale-up c4. Develop partner relationships Build a coalition c5. Train and educate stakeholders Develop educational material c7. Engage target group Increase demand Intervene with target group to enhance uptake and adherence
Author: Matthews, Andronaco and Adams Year: 2014 Aim: To assess prior conception of hazards at beaches and the impact of beach signage on the awareness of dangers among beachgoers Location: Australia	Sample: Adult beach goers (*n* = 472) at bay beaches (*n* = 2) and ocean beachers (*n* = 2) in Victoria. Recruitment: Convenience sample Response rate: 89.9%	Intervention Level: Population Type: Socio-ecological Activity: Environmental Hazard signage Duration: February–April 2012 Theory: Communication–Human Information Processing model (C-HIP) Formative research: No	Design: Quantitative In situ structured interview (quasi-experimental) Measures: Structured interviews assessing: Hazard sign identification Hazard sign recall Awareness of hazards Interpretation Ethical approval: Yes Findings: Awareness Rip current was the most reported beach hazard regardless of the presence of signage In locations where signage was present, 45% of participants reported they had noticed the signage Neither the composition of sign or hazard symbol used changed recognition or recall of warnings. Recommended there is no benefit to changing signs to incorporate four sections	Level: Target group Concepts and Strategies: c7. Engage target group Intervene with target group to enhance uptake and adherence
Author: Matthews and Franklin Year: 2018 Aim: To determine the effectiveness of a public education program, Keep Watch @ Public Pools, for improving child supervision levels by parents at public swimming pools Location: Australia	Sample: Parents and caregivers (*n* = 6930) of children aged 0–14 years (*n* = 10,186) attending public swimming pools (*n* = 7) in Melbourne Recruitment: Observational Response rate: N/R	Intervention Level: Population Type: Behavioural Activity: Environmental 6-week intervention period involved display of signage, information cards, fact sheets, training of pool lifeguards, information on Keep Watch website. Duration: 8 weeks Theory: Transtheoretical model (TTM). Formative research: No	Design: Observational In situ pre- and post-observations Measures: Supervision behaviour measured by: Attention Proximity Preparedness Child/parent ratio Ethical approval: Yes Findings: Behaviours A significant improvement in attention, proximity and preparedness was observed in parents/carers of children aged 6–10 yrs old at the intervention pools. This improvement was not observed among other age groups	Level: Provider Concepts and Strategies: c4. Develop partner relationships Promote network collaboration c5. Train and educate stakeholders Conduct educational meeting Recommendations: c1. Use evaluative and iterative strategies Assess for readiness and identify barriers and facilitators
Author: McCallin, Morgan, Camp and Yusuf Year: 2020 Aim: Assess engagement of paediatricians in drowning prevention counselling provided to families with child aged 0–10 years Location: USA	Sample: Families with children aged 0–10 yrs (*n* = 142) attending children’s hospitals (*n* = 2) in Texas Residents and paediatricians (*n* = 83) Recruitment: N/R Response rate: N/R	Intervention Level: Group Type: Behavioural Activity: Education, environmental Dissemination of community education resources, including videos, posters, safety checklists and water watcher tags Education for residents and paediatricians regarding drowning prevention Duration: 8–12 months Theory: N/R Formative research: No	Design: Quantitative Pre- and post-test Measures: Assessed provider and caregiver: Water safety counselling practice Water safety knowledge Behavioural intent Ethical approval: N/R Findings: Practice Post-intervention, 32% of residents and paediatricians reported to “always” discuss water safety behaviours with patients Knowledge All participating parents and caregivers who completed the post-test survey responded “somewhat agree” or “strongly agree” to the statement “I learnt new information about drowning prevention today from my doctor.”	Level: Provider Concepts and Strategies: c1. Use evaluative and iterative strategies Purposely re-examine the implementation
Author: McCarrison, Ren, Woomer and Cassidy Year: 2016 Aim: To evaluate an evidence-based self-instructional program called “CPR Anytime” aimed at effectively teaching paediatric CPR to parents with children enrolled in swim lessons Location: USA	Sample: Parents of children attending swimming lessons (*n* = 29) Male (*n* = 11) Female (*n* = 18) Recruitment: Convenience sample Response rate: 19.7% response rate of eligible participants; 52% retention rate at follow-up	Intervention Level: Group Type: Behavioural Activity: Testing of an existing intervention and measurement tools in a novel setting Education CPR Anytime Child program is a 20 min video-guided CPR training Duration: June–August 2015 Theory: N/R Formative research: Yes	Design: Quantitative Pre- and post-test Measures: CPR Knowledge CPR Confidence Ethical approval: Yes Findings: Knowledge and confidence Knowledge score increased from 47.3% to 93.5% post-intervention Confidence in determining need for CPR and performing CPR increased (baseline to post-intervention) Confidence in determining need also increased from baseline at 1-month follow-up Identified CPR Anytime as appropriate for pool side setting despite low uptake from parents Assessed knowledge and attitudes only, not skill	Level: Target group Concepts and Strategies: N/R c7. Engage target group Intervene with target group to enhance uptake and adherence
Author: Mitchell and Haddrill Year: 2004 Aim: Assess the methods of enforcement and evaluation of the Swimming Pools Act 1992 among NSW local councils Location: Australia	Sample: Local councils in New South Wales (*n* = 118) Rural 58% Urban 42% Recruitment: Survey emailed to all local councils in NSW Response rate: 69%	Intervention Level: Population Type: Socio-ecological Activity: Regulatory Swimming Pools Act 1992 enforces pool fencing requirements Duration: June–August 2022 Theory: N/R Formative research: Yes	Design: Quantitative Post-test only Measures: Swimming pool audit knowledge Swimming pool audit methods Compliance with legislation Ethical approval: N/R Findings: Knowledge 82% of councils were unable to report an estimated number of pools that comply with the Act Methods and compliance 28% of local councils reported understanding swimming pool inspections, 7% reported routine inspections 97% of councils provided at least educational initiative regarding pool fencing	Level: Organization, provider Concepts and Strategies: c1. Use evaluative and iterative strategies Assess for readiness and identify barriers and facilitators
Author: Mitchell and Haddrill Year: 2004 Aim: To trial a collaborative model that aimed to develop strategies to promote water safety among Chinese speakers Location: Australia	Sample: Chinese-speaking tourists and residents (*n* = 95) five focus group (*n* = 45) Intercept interviews (*n* = 50) Recruitment: Convenience and purposive sample Response rate: N/A	Intervention Level: Population Type: Behavioural Activity: Education Gaining insight of recommended education and awareness strategies from members of the Chinese advisory group Duration: 2–3 months in 2002 Theory: N/R Formative research: Yes	Design: Qualitative Focus groups Measures: Water safety-related behaviours Knowledge Attitudes towards risk communication Risk perception Understanding of signage and messaging Ethical approval: N/R Findings: Behaviour Swimming at the beach, pool or lake/river, boating and rock fishing were the most common aquatic activities Knowledge Knowledge of safety measures regarding rock fishing was low Attitudes Television campaign was the preferred method of water safety risk communication with use of idiomatic types of Chinese slogans	Level: Organization, provider, target group Concepts and Strategies: c1. Use evaluative and iterative strategies Assess for readiness and identify barriers and facilitators Conduct local needs assessment c3. Adapt and tailor context Promote adaptability Tailor strategies c4. Develop partner relationship Conduct local consensus discussions with stakeholders Promote network collaboration Use advisory boards and workgroups Recommendations: c5. Train and educate stakeholders Conduct educational meetings
Author: Moran and Stanley Year: 2006 Aim: To evaluate a pilot parent education program aimed at enhancing knowledge and attitudes Location: New Zealand	Sample: Parents of children aged 2–4 yrs enrolled in swimming lessons (*n* = 106) at swim schools in Auckland (*n* = 2) Recruitment: N/R Response rate: N/R	Intervention Level: Group Type: Behavioural Activity: Education 10-week informal water safety program in conjunction with swim school lessons Duration: 10 weeks Theory: N/R Formative research: Yes	Design: Quantitative Pre- and post-test Measures: Knowledge Attitudes and beliefs Ethical approval: N/R Findings: Knowledge Post-program, fewer parents agreed with misconceptions regarding water safety around children Child CPR knowledge did not improve post-intervention Attitudes Measures of attitudes towards water safety overall improved from 17% to 54% post-program	Level: Organization Concepts and Strategies: c1. Use evaluative and iterative strategies Stage implementation scale up
Author: Moran, Stanley and Rutherford Year: 2012 Aim: To evaluate a parent education program aimed at enhancing CPR knowledge and confidence Location: New Zealand	Sample: Parents of children under 5 yrs (*n* = 109) enrolled in swim lessons at swim centres in Auckland (*n* = 4) Control group (*n* = 36) Pool-based instruction (*n* = 37) Home-based instruction (*n* = 36) Male 14% Female 86% Recruitment: N/R Response rate: 76%	Intervention Level: Group, individual Type: Behavioural Activity: Education Pool-based instruction Training by CPR instruction in conjunction with swim lessons Home-based instruction Provided take-home kit with CPR manikin, DVD, skills checklist, etc. Duration: N/R Theory: N/R Formative research: No	Design: Quantitative Pre- and post-test; quasi-experimental Measures: CPR confidence CPR knowledge Ethical approval: Yes Findings: Knowledge The proportion of correct responses regarding CPR significantly improved among both interventions Confidence in ability to perform CPR significantly increased among both pool-based and home-based groups There were no significant differences between the two intervention groups regarding knowledge and confidence of performing CPR.	Level: Organization, provider, target group Concepts and Strategies: c1. Use evaluative and iterative strategies Assess for readiness an identify barriers and facilitators Conduct local needs assessment c2. Provide interactive assistance Facilitation c4. Develop partner relationships Develop academic partnerships c6. Support providers Develop resource sharing agreements c7. Engage target group Intervene with target group to enhance uptake and adherence Involve target group and support network Recommendations: c4. Develop partner relationships Build a coalition c5. Train and educate stakeholders Use train-the-trainer strategies
Author: Moran, Webber and Stanley Year: 2017 Aim: To evaluate the 4Rs of Aquatic Rescue by measuring the uptake of rescue information and emergency procedures Location: New Zealand	Sample: Parents/caregivers of children enrolled in water safety lessons (*n* = 467) at Auckland swim schools (*n* = 8) Male 23% Female 77% Recruitment: Convenience sample Response rate: 37% of enrolled children participated in the survey	Intervention Level: Group Type: Behavioural Activity: Education 5-day in-water program. Resources Dissemination of pamphlets, webpage with downloadable resources, video, newspaper release Study informed by authors’ previous research Duration: Approx. 4 months Theory: N/R Formative research: No	Design: Quantitative Pre- and post-test Measures: Skills (swim competency, CPR) Knowledge Confidence Perception of risk Ethical approval: N/R Findings: Knowledge Bystander rescue knowledge improved Confidence 56% reported “they would be too afraid” to conduct a bystander rescue post-intervention Knowledge of how to perform a rescue did not impact confidence	Limited detail of implementation other than it was developed based on need (not described) and previous interventions from Canada and Australia. Level: N/R Concepts and Strategies: N/R
Author: Morrongiello, Sandomierski, Schwebel and Hagel Year: 2013 Aim: To explore whether parents with children enrolled in swim lessons, beliefs, judgements and perceptions are impacted by regular feedback from the instructor Location: Canada	Sample: Parents of children aged 2–5 yrs enrolled in swim lessons (*n* = 387) Intervention group: 16% Control group: 84% Male: 15.5% Female: 84.5% Recruitment: N/R Response rate: Consent rate: 23%	Intervention Level: Group Type: Behavioural Activity: Education Control (Swim/only) Intervention (Swim/parent): involved regular written feedback from the swim instructor regarding the child’s progress Duration: 10 weeks Theory: N/R Formative research: No	Design: Quantitative Pre- and post-test Measures: Parent and instructor surveys: Beliefs regarding supervision needs and ability for child to keep themselves safe from drowning Parental and instructor estimation of child’s swimming ability Ethical approval: Yes Findings: Beliefs There were no significant differences regarding drowning prevention and supervision needs between intervention and control groups Accuracy of parental estimation of child’s swimming abilities remained poor throughout	Level: Organization, target group Concepts and Strategies: c4. Develop partner relationships Develop academic partnerships
Author: Morrongiello, Sandomiersk, and Spence Year: 2014 Aim: To determine how children’s participation in swim lessons impacts parents’ appraisals of children’s drowning risk and need for supervision Location: Canada	Sample: Parents of children aged 2–5 yrs enrolled in swim lessons (*n* = 387) Male: 15% Female: 85% Recruitment: Convenience and purposive Response rate: 69% participant attrition from T1–T4	Intervention Level: Group Type: Behavioural Activity: Educational Swimming lessons (1 lesson × 10 weeks) Duration: 8 months Theory: N/R Formative research: No	Design: Quantitative Pre- and post-test (four wave) Measures: Perceptions of: Child’s swim ability Supervision needs Child’s ability to keep themselves from drowning Ethical approval: Yes Findings: Attitudes As children progressed through swim lessons, perceptions of swim ability and ability to keep themselves safe increased High perception scores of children’s ability to keep themselves from drowning predicted a reduction in ratings of children’s supervision needs belief score	Level: Organization, target group Concepts and Strategies: c1. Use evaluative and iterative strategies Conduct local needs assessment c4. Develop partner relationships Develop academic partnerships
Author: Olaisen, Flocke and Love Year: 2018 Aim: To measure the relationship of age, gender and number of swimming lessons on skill acquisition by Latino immigrant children Location: USA	Sample: Low-income, Latino children aged 3–14 yrs (*n* = 312) in California Male (*n* = 83) Female (*n* = 66) Recruitment: Purposive Response rate: Of the eligible participants, 52.2% of participants were complete cases	Intervention Level: Group Type: Behavioural Activity: Education Learn-to-swim community program Participants were assigned groups specific to age and swimming ability Duration: 8 weeks Theory: Health Belief Model and socio-ecological framework Formative research: No	Design: Observational In situ observation (all lessons) Measures: Observed swim skills Gender Age Behaviour Confidence Ethical approval: Yes Findings: Skills Average swim skill acquisition improved A difference in skill acquisition was found between participants that completed <7 lessons and participants that completed >10	Level: Organization, provider, target group Concepts and Strategies: c3. Adapt and tailor context Promote adaptability c4. Develop partner relationships Promote network collaboration Use advisory boards and workgroups c5. Train and educate stakeholders Conduct educational meetings Making training dynamic c7. Engage target group Increase demand Intervene with target group to enhance uptake and adherence
Author: Olivar Year: 2019 Aim: To explore the impact cognitive and creative teaching styles have on improving aquatic competences and perceptions Location: Uraguay	Sample: Children aged 10–12 yrs (*n* = 74) in Montevideo Recruitment: N/R Response Rate: N/R	Intervention Level: Group Type: Mixed Activity: Education Junior lifeguard program offered during school hours as a part of the Everyone, enjoy the water intervention 2 lessons/week for 16–24 lessons Duration: 8–12 weeks Theory: Ecological motor and sport learning (environment, student and demands of the task) Formative research: No	Design: Observational In situ observation Measures: Swim skills Water safety skills Attitudes Ethical approval: N/R Findings: Results not clearly outlined	Level: Provider, target group Concepts and Strategies: c1. Use evaluative and iterative strategies Conduct cyclical small tests of change Purposely re-examine the implementation c2. Provide interactive assistance Facilitation c3. Adapt and tailor context Promote adaptability c7. Engage target group Increase demand Intervene with target group to enhance uptake and adherence
Author: Petrass and Blitvich Year: 2014 Aim: To assess the impact of a 12-week intervention on swimming survival and rescue skills, water safety knowledge and attitudes Location: Australia	Sample: First year Bachelor of Exercise and Sports Science (*n* = 135) in Victoria Male 56.4% Female 43.6% Recruitment: Convenience sample Response Rate: Students enrolled (*n* = 154)	Intervention Level: Group Type: Behavioural Activity: Education Can You Swim Intervention Involved a 12-week intervention encompassing survival and rescue skills and water safety knowledge 1 × 1 h practical session/week 1 × 1 h theory session/week Duration: 5 months Theory: Social learning pedagogy Formative research: No	Design: Quantitative Pre- and post-test Measures: Knowledge Attitudes Swimming abilities Ethical approval: Yes Findings: There was no change in attitudes Knowledge There was a significant improvement in water safety knowledge post-intervention Skills Practical testing found improvement in overall swimming ability post-intervention	Level: Organization, provider Concepts and Strategies: c1. Use evaluative and iterative strategies Conduct local needs assessment c3. Adapt and tailor context Promote adaptability c7. Engage target group Increase demand
Author: Petrass, Simpson, Blitvich, Birch and Matthews Year: 2021 Aim: To examine whether participation in a survival swim program resulted in improved aquatic skills and knowledge among year 5 and 6 primary school students Location: Australia	Sample: Grade 5 and 6 primary school students (*n* = 204) from regional (*n* = 2) and metropolitan schools (*n* = 1). Regional (*n* = 68) Metropolitan (*n* = 111) Male 48.6% Female 51.4% Recruitment: N/R Response rate: N/R	Intervention Level: Group Type: Behavioural Activity: Education Student-centred aquatic program 10 × 1 h sessions Duration: 10 weeks Theory: N/R (states use of theory but theory name not recorded) Formative research: No	Design: Mixed methods Pre- and post-test (survey and practical skills assessment) Measures: Water safety knowledge Perceived swimming competency Practical aquatic skills Ethical approval: Yes Findings: Skills Both regional and metropolitan groups showed improvements in aquatic skills practical testing Knowledge There was no change in knowledge scores post-intervention among the regional students. Among the metropolitan students, there was a significant improvement in knowledge scores	Level: Organization, provider, target group Concepts and Strategies: c1. Use evaluative and iterative strategies Assess for readiness and identify barriers and facilitators Develop and implement tools for quality-monitoring Purposely re-examine the implementation c4. Develop partner relationships Involve executive boards c5. Train and educate stakeholders Conduct educational meetings Develop educational materials Distribute educational materials c7. Involve target group Involve target group and support networks
Author: Quan, Shephard and Bennett Year: 2020 Aim: To describe the development, components and evaluation of a drowning prevention campaign for the Vietnamese–American community conducted in 2006–2007 Location: USA	Sample: Vietnamese parents within the community Pre-intervention (*n* = 168) Male 27% Female 73% Post-intervention (*n* = 230) Male 38% Female 62% Recruitment: Convenience sample Response rate: N/R	Intervention Level: Population Type: Mixed Activity: Environment Implement lifeguarding at aquatic locations Lifejacket loaner board program Recruit Vietnamese/Asian lifeguards Behavioural Community presentations and engagement Poster and brochure dissemination Media release Free/low-cost swimming lessons Duration: Pre-intervention surveys: Dec 2006, Mar 2007 Post-intervention surveys: Nov 2007 and Apr 2008 Theory: Campaign development used PRECEED-PROCEED framework; Social Marketing Formative research: Yes	Design: Quantitative Pre- and post-test Measures: Swim lesson status Lifejacket use Swimming behaviours Campaign awareness Lifeguarded aquatic parks Status lifejacket loaner program Ethical approval: Yes Findings: Awareness There was a statistically significant increase in awareness of water safety information Behaviours There was not a significant increase in lifejacket use and attendance of swimming lessons for children Environment Availability of swim lessons and low-cost swim lessons increased. The availability of free lifejackets increased	Level: Organization, target group Concepts and Strategies: c1. Use evaluative and iterative strategies Conduct local needs assessment c4. Develop partner relationships Conduct local consensus discussions Promote network collaboration c6. Support providers Create new provider teams c7. Engage target group Intervene with target group to enhance uptake and adherence Use mass media c8. Financial strategies Alter fees for target group
Author: Rawlins Year: 2018 Aim: To describe the Historically Black Colleges or Universities’ (HBCUs) efforts to overcome barriers to water competence and provide water safety education in African American communities Location: USA	Sample: African American members of the Delaware community; parents (*n* = 25) students (*n* = 38) Recruitment: N/R Response rate: N/R	Intervention Level: Group Type: Behavioural Activity: Education Parent orientation to swim lessons Learn to swim initiative Duration: Orientation held in January and March 2017 Theory: N/R Formative research: Focus group conducted although did not seem to inform strategies/study	Design: Process evaluation Measures: Attendance Ethical approval: N/R Findings: Behaviours 20% of participants in Parent Orientation to Swim Lessons signed up for formal swim lessons	Level: Organization, providers, target group Concepts and Strategies: c1. Use evaluative and iterative strategies Conduct local needs assessment Stage implementation scale-up c4. Develop partner relationships Build a coalition Promote network collaboration c6. Support providers Create new provider teams
Author: Sandomierski, Morrongiello and Colwell Year: 2019 Aim: To develop, implement and evaluate the S.A.F.E.R. Near Water Program Location: Canada	Sample: Parents of children aged 2–5 yrs enrolled in swimming lessons (*n* = 242) at public (*n* = 2) and private (*n* = 2) swim organizations in Ontario Intervention (*n* = 92) Male 28% Female 72% Control (*n* = 150) Male 19% Female 81% Recruitment: Convenience sample Response rate: Final participation rate for the intervention group was 42% and 55% for the control group	Intervention Level: Group Type: Behavioural Activity: Education S.A.F.E.R. Near Water Program 2 × 30 min in person seminars Videos Poster display Duration: N/R Theory: Health Belief Model, Theory of Planned Behaviour and Protection Motivation Theory Formative research: No	Design: Quantitative Pre- and post-test Quasi- experimental Measures: Knowledge Behavioural intentions Perceptions of supervision Beliefs Ethical approval: Yes Findings: Behavioural intent Intervention participants more likely to intend closer supervision of children than control group Knowledge and attitudes Increase is knowledge of drowning risk and need for supervision in intervention group Greater optimism bias and inaccuracy of judgement of children’s swim skills in control group Feasibility of providing parent education during children’s swimming lessons	Level: Organization, provider Concepts and Strategies: c4. Develop partner relationships Develop academic partnerships c6: Support providers Develop resource sharing agreements c7. Engage target group Intervene with target group to enhance uptake and adherence Recommendations: c4. Develop partner relationships Promote network collaboration
Author: Savage and Franklin Year: 2015 Aim: To increase understanding of aquatic related programs and the needs of CALD communities in NSW Evaluate AUSTSWIM’s current training methods in relation to the training of CALD candidates Location: Australia	Sample: Aquatic facilities (*n* = 51); regional (*n* = 29); metropolitan (*n* = 23) Focus group (*n* = 15) included swim teachers, aquatic facility representatives, CALD AUSTSWIM candidates, community leaders, AUSTSWIM course participants (*n* = 63) Recruitment: Convenience Response rate: 10% aquatic facilities	Intervention Level: Group Type: Behavioural Activity: Education AUSTSWIM Teacher of Swimming and Water Safety course (2 days, 8 h/day) Duration: Approx. 6 months Theory: N/R Formative research: Study is partly formative research.	Design: Mixed methods Post-test only (survey) Focus groups Measures: Swim course completion CALD program status Perceptions of water safety Ethical approval: Yes Findings: Programs 19/51 facilities reported offering CALD-specific programs Skills 52% candidates completed all swimming and water safety teacher requirements	Level: Organization, provider Concepts and Strategies: c1. Use evaluative and iterative strategies Conduct local needs assessment c2. Provide interactive assistance Facilitation c4. Develop partner relationships Conduct local consensus discussions Develop academic partnerships Recommendations: c7. Engage the target group Increase demand Intervene with target group to enhance uptake and adherence
Author: Scurati, Michielon, Signorini and Invernizzi Year: 2019 Aim: To assess the effects of different teaching methods on breaststroke skill development Location: Italy	Sample: Sport science students (*n* = 16) Recruitment: Convenience Response rate: N/R	Intervention Level: Group, individual Type: Behavioural Activity: Education Learn-to-swim 8 × 45 min/week Control: Ordinary lessons Intervention: Utilised mobile device support Duration: 8 weeks Theory: Learner-centred pedagogy and reflection-actions process Formative research: No	Design: Observational In situ and video observation Measures: Swimming skills Technique Ethical approval: Yes Findings: Skills Improvements in swimming skills were comparable among the control and intervention groups	Level: Organization Concepts and Strategies: c4. Develop partner relationships Develop academic partnerships
Author: Spitzer, Mangione, Chow, Quan and Bennett Year: 2019 Aim: To assess behaviours and perceptions regarding lifejacket use and loaner board programs among parents Location: USA	Sample: Parents (*n* = 102) at swim sites (*n* = 10) Control sites (*n* = 7) Intervention sites (*n* = 3) Recruitment: Convenience Response rate: N/R	Intervention Level: Population Type: Socio-ecological Activity: Loaner board program Duration: 2 months Theory: Social Marketing Theory Formative research: Yes	Design: Mixed methods Parent surveys Observational study Measures: Behaviours: Flotation device type use Awareness of program Strategy suggestions Ethical approval: Yes Findings: Awareness Where loaner boards were present, 45% of parents were aware of facility before arriving Behaviours 68% of families brought a flotation device with intended use for children Perception The most common suggestion to increase lifejacket use was to increase loaner board availability (32%)	Level: Organization Concepts and Strategies: c4. Develop partner relationships Develop academic partnerships
Author: Stempski, Liu, Grow, Pomietto, Chung, Shumann and Bennett Year: 2015 Aim: To conduct process evaluation of the Everyone Swims community partnership to identify policy and system changes and understand their potential impact Location: USA	Sample: King County, Washington Organizations (*n* = 14) representing community health clinic sites (*n* = 21) Pools (*n* = 28) Beaches (*n* = 9) Rowing houses (*n* = 2) Focus group participants (*n* = 51) Black, White, Latino, Vietnamese and Somali families Recruitment: N/R Response rate: 100%	Intervention Level: Population Type: Socio-ecological Activity: Development of the community partnership Everyone Swims Focus groups (*n* = 5) with members of priority groups Duration: 18 months Theory: Socio-ecological model Formative research: No	Design: Process evaluation Measures: System and policy changes Barriers Achievements Feedback Ethical approval: Yes Findings: Policy 79% of outlined policy changes were implemented by one or more partners (improvements to systems, referral practices, scholarship accessibly and increased availability of swim programs).	Level: Organization, provider Concepts and Strategies: c1. Use evaluative and iterative strategies Conduct local needs assessment c2. Provide interactive assistance Provide technical assistance c3. Adapt and tailor context Promote adaptability c4. Develop partner relationships Capture and share local knowledge Conduct local consensus discussions Develop academic partnerships Identify and prepare champions Obtain formal commitments Organise implementation team meetings Promote network collaboration c8. Financial strategies Alter provider incentives/allowance structures
Author: Terzidis, Koutroumpa, Skalkidis, Matzavakis, Malliori, Frangakis, DiScala and Petridou Year: 2007 Aim: To explore whether an intervention during school can lead to changes in water safety knowledge and attitudes Location: Greece	Sample: School-aged children in Greater Athens Kindergarten/grade one Intervention (*n* = 115) Control (*n* = 202) Elementary Intervention (*n* = 205) Control (*n* = 220) High school Intervention (*n* = 321) Control (*n* = 337) Recruitment: Convenience sample Response rate: N/R	Intervention Level: Group Type: Behavioural Activity: Education School-based intervention with: Audio-visual presentations Discussion of personal experiences Take-home materials Duration: 1 month Theory: N/R Formative research: Yes	Design: Quantitative Pre- and post-test Measures: Knowledge Attitudes Ethical approval: Yes Findings: Knowledge and attitudes The intervention was effective in improving knowledge (17.4%) and attitudes (23.64%) mean scores among the kindergarten and grade one group. This change was not found among the elementary and high school groups	Level: Organization, provider Concepts and Strategies: c3. Adapt and tailor context Promote adaptability Tailor strategies c4. Develop partner relationships Promote network collaboration c7. Engage target group Increase demand
Author: van Weerdenburg, Mitchell and Wallner Year: 2006 Aim: To describe compliance levels with the Swimming Pools Act 1992 and identify perceived barriers to effective management among three local councils To describe pool owners’ attitudes to pool fencing and inspections Location: Australia	Sample Pool safety compliance registers within local councils (*n* = 3) in NSW Interviews with local council staff (*n* = 3) Survey Pool owners residing in council A (*n* = 205). Recruitment: Purposive/Convenience Response rate: 20.4% of pool owners residing in council A responded to the survey	Intervention Level: Population Type: Socio-ecological Activity: Regulation Councils in NSW to appropriately action and enforce pool fencing requirements Duration: N/R Theory: N/R Formative research: No	Design: Mixed methods Audit of record keeping Interviews (council staff) Surveys (pool owners) Measures: Audit Database, register, staff roles, Act enforcement, contact with pool owners, non-compliance management Interviews Barriers to effective management Key issues Surveys Attitudes—pool fencing Behaviours Ethical approval: N/R Findings: Attitudes 95.6% of pool owners in council A supported pool fencing requirements and inspections Inconsistency and misinterpretation of the act was outlined as a barrier to enforcement among council employees and lack of funding Elected councilors were discussed as a barrier and facilitator to pool fencing inspection practices Regulation behaviours Compliance with pool fencing legislation in councils with minimal inspection activity was poor	Level: Organization, provider, target group Concepts and Strategies: c1. Use evaluative and iterative strategies Assess for readiness and identify barriers and facilitators Develop and implement tools for quality-monitoring c8. Financial strategies Develop disincentives c9. Change infrastructure Change liability laws or enforcement Recommendations: c4. Develop partner relationships Inform local opinion leaders c5. Train and educate stakeholders Conduct educational meetings c7. Engage target group Intervene with target group to enhance uptake and adherence c8. Financial strategies Place evidence-informed interventions on a fee-for-service list/formularies c9. Change infrastructure Change liability laws or enforcement
Author: Wilks, Kanasa, Pendergast and Clark Year: 2017 Aim: To assess beach safety awareness among a group of primary school students before and after a one-day training program. Pilot evaluation of key beach safety topics and development of new evidence-based learnings Location: Australia	Sample: Year 6 students attending a private school in South-East Queensland (*n* = 107) Recruitment: Convenience sample (1 year group from 1 school) Response rate: *n* = 107 recruited *n* = 102 post survey *n* = 105 8 week follow-up	Intervention Level: Group Type: Behavioural Activity: Curriculum developed specialists and expanded on the QLD Health Beach Safe Schools Program 1-day interactive session delivered by lifeguards (*n* = 9) during school hours Beach safety (flags and signage, lifeguard identification and role, rip currents) Leadership and team bonding exercises First aid and CPR training Workbook completion Participation certificate Duration: 8 weeks (1 day intervention) Theory: N/R Formative research: Yes	Design: Quantitative Pre- and post-test (three wave) Measures: 50-item quiz Water safety knowledge Rip identification Hazard signage recognition Lifeguard identification Published literature used to develop the quiz Content validity undertaken with ten Surf Life Saving QLD members. Ethical approval: Yes Findings: Knowledge Participants reported improvement in ability to define and identify a rip Hazard signage with low recognition pre-intervention significantly improved	Level: Organization, provider, target group Concepts and Strategies: c1. Use evaluative and iterative strategies Conduct local needs assessment Stage implementation scale-up c3. Adapt and tailor context Tailor strategies c4. Develop partner relationships Promote network collaboration. c5. Train and educate stakeholders Work with educational institutions
Author: Yusuf, Jones, Camp and McCallin Year: 2022 Aim: To increase drowning prevention counselling provided during paediatric visits to children aged 0–10 yrs and to evaluate the attainment of drowning prevention knowledge by physicians and caregivers Location: USA	Sample: Texas physicians (*n* = 33) in primary health care office (*n* = 23), urgent care (*n* = 6) or emergency department (*n* = 4). T1: *n* = 10; T2: *n* = 23 Each physician counselled 60 caregivers (*n* = 1934) T1 (*n* = 600); T2 (*n* = 1334) Recruitment: Via the project co-leaders at two academic affiliated institutions or communications from state paediatric society Response rate: N/R	Intervention Level: Group Type: Mixed Activity: Developed and implemented a water safety counselling program for paediatricians to impart to families during well-child, urgent care and ED visits for children aged 0–10 years Physicians: Introductory webinar: Power Point presentation Video exemplar Project implementation guide Caregivers 1–4 min clinical encounter Educational resources Duration: 2018 (T1) and 2019 (T2) each lasting 2–3 months Theory: N/R Formative research: No	Design: Quantitative Pre- and post-test (physicians) Post-test only (caregivers) Measures: Physicians Counselling rate Process evaluation Both physicians and caregivers were surveyed on evidence-based drowning prevention strategies: Four-sided fencing of pools (T2), Touch supervision (T1 and T2), Life jackets (T2), Swim/CPR classes (T2) Ethical approval: No (according to authors) Findings: Physicians demonstrated improvement in discussing drowning prevention in office setting. Paediatricians had adequate drowning prevention knowledge; an efficient counselling strategy helped them impart this knowledge to their patients Increased drowning prevention counselling during well-child visits. Counselling frequency increased from 54% to 70% from T1 to T2 100% of physicians correctly identified the best drowning prevention strategy Caregiver acquisition of new water safety information improved from T1 (6808%) to T2 (80.6%)	Level: Provider Concepts and Strategies: c1. Use evaluative and iterative strategies Develop and implement tools for quality-monitoring Obtain and use target group and support network feedback. Purposely re-examine the implementation. c5. Train and educate stakeholders Create a learning collaborative Work with educational institutions

Abbreviations: CPR—cardiopulmonary resuscitation, LJ—life jacket, PFD—personal flotation device, N/A—not applicable, N/R—not recorded, USA—United States of America.

**Table 4 ijerph-21-00045-t004:** Summary of implementation concepts and strategies identified in the drowning prevention literature.

KEY						
	Key strategy (identified ≥ 5 times)			Identified ≤ 4 times		Not identified
Identified		Concept and Strategy	*n* *	Examples		Articles
		Develop academic partnership (c4)	32			
		Develop academic partnerships (c4_04)	18	Ethics Board approvals cited. University input in content or resource development. Identified collaboration and consultation with researchers.		[46,50,51,52,53,54,60,61,64,65,67,75,77,78,86,87,88,89]
		Promote network collaboration (c4_13)	14	Program staff from various backgrounds and stakeholder groups Working with not-for-profit organizations Providing opportunities for stakeholder collaboration Expert input into campaign and evaluation tools Multi-sectorial collaboration and community partnerships		[48,49,52,59,61,69,73,79,83,84,85,89,90,92]
		Build a coalition (c4_01)	8	Description of collaboration and partnerships and various roles of organizations throughout the intervention		[46,48,50,63,64,67,75,84]
		Conduct local consensus discussions (c4_03)	5	Interaction with stakeholders about the intervention Identification of ability of stakeholders or facilities to undertake the intervention (conducted with stakeholders)		[56,73,83,86,89]
		Use advisory boards and workgroups (c4_15)	3			
		Inform local opinion leaders (c4_08)	2			
		Capture and share local knowledge (c4_02)	2			
		Identify and prepare champions (c4_06)	1			
		Obtain formal commitments (c4_11)	1			
		Organise implementation team meetings (c4_12)	1			
		Develop an implementation glossary (c4_05)	0			
		Identify early adopters (c4_07)	0			
		Model and simulate change (c4_10)	0			
		Recruit, designate and train for leadership (c4_14)	0			
		Use an implementation advisor (c4_16)	0			
		Visit other sites (c4_17)	0			
		Use of evaluative and iterative strategies (c1)	29			
		Conduct local needs assessment (c1_04)	11	Understanding of target group and support network needs via focus groups, survey data collection, literature review, mapping of local waterway use and drowning trends		[56,64,73,75,77,81,83,84,86,89,92]
		Assess for readiness and identify barriers and facilitators (c1_01)	9	Assessing barriers for time, materials and transportation Surveying current pool fencing compliance Testing workforce and development needs of CALD communities Reviewing swim school staff’s CPR qualifications		[50,56,57,72,73,75,82,91,93]
		Stage implementation scale-up (c1_10)	8	Description of pilot intervention Collection of baseline data for future observational studies Identify issues of scaling up too quickly		[50,55,56,67,74,84,90,92]
		Develop and implement tools for quality-monitoring (c1_06)	7	Tools developed to evaluate the intervention Observation strategies put in place Study protocols documenting all changes and revisions to the intervention		[53,54,55,56,59,82,91]
		Purposely re-examine the implementation (c1_09)	2			
		Cyclical small tests of change (c1_03)	2			
		Obtain and use target group and support network feedback (c1_08)	2			
		Audit and provide feedback (c1_02)	0			
		Develop a formal implementation blueprint (c1_05)	0			
		Develop and organize quality-monitoring systems (c1_07)	0			
		Engage the target group (c7)	22			
		Intervene with target group to enhance uptake and adherence (c7_02)	14	Involve target group in intervention development or implementation Telephone interviews to assess uptake and adherence Barriers such as language and childcare resolved Ensuring participants comfort as intervention progresses Community ownership of intervention messaging Location and timing of intervention		[47,53,55,65,67,68,71,75,79,80,83,85,86,91]
		Increase demand (c7_01)	7	Use of existing events and networks to reach the target group at a convenient time or location		[57,58,67,80,81,86,90]
		Use mass media (c7_05)	6	Use of mass media to informe the community of intervention messages		[52,53,54,60,62,83]
		Involve the target group and support network (c7_03)	4			
		Prepare target group to actively participate (c7_04)	0			
		Adapt and tailor context (c3)	13			
		Promote adaptability (c3_01)	9	Assessment of suitability of an existing program Community involvement in adapting an existing program Adaptability in levels of involvement in activities Priority variations for different intervention locations		[45,50,56,73,79,80,81,89,90]
		Tailor strategies (c3_02)	8	Swim teaching techniques tailored to each participant Barriers such as facility availability, language, travel time, transportation, local water ways and weather taken into account		[45,48,50,64,73,85,90,92]
		*Use data experts (c3_03)*	0			
		*Use data warehouse techniques (c3_04)*	0			
		Train and educate stakeholders (c5)	12			
		Conduct educational meetings (c5_01)	6	Training or meeting with community leaders, providers and stakeholders to share information about the intervention prior to delivering it to the target group Training varied in length and content		[66,69,73,79,81,91]
		Work with educational institutions (c5_11)	3			
		Develop educational materials (c5_05)	2			
		Create a learning collaborative (c5_04)	1			
		Distribute educational materials (c5_06)	1			
		Make training dynamic (c5_07)	1			
		Use train-the-trainer strategies (c5_10)	1			
		Conduct educational outreach visits (c5_02)	0			
		Conduct ongoing training (c5_03)	0			
		Provide ongoing consultation (c5_08)	0			
		Shadow other experts (c5_09)	0			
		Provide interactive assistance (c2)	6			
		Facilitation (c2_02)	5	Interactive problem solving undertaken with the target group, their support network and/or providers		[45,48,75,80,86]
		Provide technical assistance (c2_03)	1			
		*Centralise technical assistance (c2_01)*	0			
		Provide supervision (c2_04)	0			
		Support providers (c6)	6			
		Develop resource-sharing agreements (c6_02)	4			
		Create new provider teams (c6_01)	3			
		Facilitate relay of program data to providers (c6_03)	0			
		Remind providers (c6_04)	0			
		Revise professional roles (c6_05)	0			
		Financial strategies (c8)	5			
		Fund and contract for the evidence-informed intervention (c8_05)	2			
		Alter provider incentives/allowance structures (c8_02)	1			
		Alter fees for target group (c8_03)	1			
		Develop disincentives (c8_04)	1			
		Place interventions on a fee-for-service list/formularies (c8_07)	1			
		Access new funding (c8_01)	0			
		*Make billing easier (c8_06)*	0			
		*Use capitated payments (c8_08)*	0			
		*Use other payment schemes (c8_09)*	0			
		Change infrastructure (c9)	4			
		Change liability laws or enforcement (c9_02)	2			
		Change record systems (c9_04)	2			
		Change service sites (c9_05)	1			
		*Change accreditation or membership requirements (c9_01)*	0			
		Change physical structure and equipment (c9_03)	0			
		*Create or change credentialing and or licensure standards (c9_06)*	0			
		Mandate change (c9_07)	0			
		Start a dissemination organization (c9_08)	0			

* number of articles where the strategy was identified.

## Data Availability

No new data were created or analyzed in this study. Data sharing is not applicable to this article.

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
