# Peer review of "Using ERIC to Assess Implementation Science in Drowning Prevention Interventions in High-Income Countries: A Systematic Review"

_ijerph, 2023, doi:10.3390/ijerph21010045_

Round 1
Reviewer 1 Report
Comments and Suggestions for Authors
Thank you for the opportunity to review your good work in drowning prevention interventions.
Title:
Consider shortening the title by removing the first word "Who" and start with Using Expert Recommendation for Implementing Change (ERIC) in Drowning Prevention. Implementation Science in high income countries: A Systematic Review.
Introduction:
Consider removing the word "however" to start the 2nd sentence in line 10.
Consider removing "Firstly even" and begin with If or When interventions --- in line 15 and "Secondly" and begin with Intervention implementation -- in line 17.
Line 30 - consider removing the word "Whilst" and start that sentence with "This tool ---.
Line 42-45 - consider changing the word "whilst" with while. (x 3 in that sentence)
Line 70 - consider removing the start of this sentence "With the recent call to action in mind" and start this sentence with This systematic review---
Materials and Methods:
This section is very well done.
Results:
This section is very well done.
Overview of case studies: (interventions and evaluations)
This section is very well done.
Implementation key concepts and strategies:
This section is very well done.
Discussion:
This section is very well done.
Understanding the use of implementation strategies in drowning prevention interventions:
Line 377 - consider - This aligns with the way drowning prevention is framed ---
Line 396 - consider a new paragraph - The review-----
Line 406 - consider replacing "whilst" with while or eliminating this link word and continuing after the colon with developing---
Gaps in use and reporting of implementation strategies:
Line 482 - consider changing this sub-title to something like - Lessons learned or What was learned or Learning highligts or The learning:
Strengths and limitations:
Consider making a comment/sentence about the Australia, USA dominance of the literature from HICs. Does this represent funding availability or academic partnership availability or ??? I also note that non-english articles - was only 6. That issue may also be linked to funding and academic availability.
Conclusions:
This is well done.
Consider adding somewhere that is most appropriate that implementation research is most needed in LMICs where drowning rates and the impact of interventions is larger.
This paper is of very high value to the drowning prevention community and the drowning prevention effort!!
Thank you!
Comments on the Quality of English Language
very minor edits suggested for consideration
Author Response
RESPONSE TO REVIEWER 1
Thank you for taking the time to review the manuscript ‘Who’s using ERIC in drowning prevention? Implementation science in drowning prevention interventions in high-income countries: A Systematic Review’ and for your considered feedback. We have responded to your comments and believe this has helped improve the quality of the manuscript.
Comment 1
Title:
Consider shortening the title by removing the first word "Who" and start with Using Expert Recommendation for Implementing Change (ERIC) in Drowning Prevention. Implementation Science in high income countries: A Systematic Review.
Response 1: Thank you for your feedback The title has been altered as you suggested. The change has been highlighted in yellow in the manuscript.
Comment 2
Introduction:
Consider removing the word "however" to start the 2nd sentence in line 10.
Consider removing "Firstly even" and begin with If or When interventions --- in line 15 and "Secondly" and begin with Intervention implementation -- in line 17.
Line 30 - consider removing the word "Whilst" and start that sentence with "This tool ---.
Line 42-45 - consider changing the word "whilst" with while. (x 3 in that sentence). This sentence included typographical errors that have now been corrected. It now reads: In low and middle income countries (LMICs) there is a higher prevalence of children drowning close to home due to issues of supervision, barriers to water sources and water safety skills, while older children and adults drown when undertaking work or during travel on water. The altered text has been highlighted in yellow.
Line 70 - consider removing the start of this sentence "With the recent call to action in mind" and start this sentence with This systematic review-.
Response 2: Thank you for your suggestions above. The changes recommended have all been made as you suggested and are highlighted in yellow.
Materials and Methods:
This section is very well done.
Response: Thank-you for your positive feed back
Results:
This section is very well done.
Response: Many thanks again for the positive comments
Overview of case studies: (interventions and evaluations)
This section is very well done.
Thank-you.
Implementation key concepts and strategies:
This section is very well done.
Response: Thank-you.
Discussion:
This section is very well done.
Response: Many thanks.
Comment 3
Understanding the use of implementation strategies in drowning prevention interventions:
Line 377 - consider - This aligns with the way drowning prevention is framed ---
Line 396 - consider a new paragraph - The review-----
Line 406 - consider replacing "whilst" with while or eliminating this link word and continuing after the colon with developing---
Response 3: Thank you for your feedback above. The changes recommended have been made as you suggested. The changes in the manuscript are highlighted in yellow.
Comment 4
Gaps in use and reporting of implementation strategies:
Line 482 - consider changing this sub-title to something like - Lessons learned or What was learned or Learning highlights or The learning:
Response 4: Thank you for your feedback. The subheading has been altered to “What was learnt?” and has been highlighted in the manuscript in yellow.
Comment 5
Strengths and limitations:
Consider making a comment/sentence about the Australia, USA dominance of the literature from HICs. Does this represent funding availability or academic partnership availability or ??? I also note that non-english articles - was only 6. That issue may also be linked to funding and academic availability.
Response 5: Thank you for your feedback. A sentence has been added about the over-representation of literature from Australia and USA. The following text has been added and is highlighted in the manuscript in yello: An over representation of literature from Australia and the USA may reflect drowning prevention efforts have attracted funding and resources which has allowed for a research-practice nexus to be established and afforded peer review publications. In contrast there were few non-English articles suggesting drowning prevention may be a lower priority for research funding in some countries and the opportunity to publish becomes limited.
Conclusions:
This is well done.
Response: Thank you!
Comment 6
Consider adding somewhere that is most appropriate that implementation research is most needed in LMICs where drowning rates and the impact of interventions is larger.
Response 6: After careful consideration we feel this is beyond the scope of this paper as the research was into HICs only. We respectfully have chosen to omit this statement.
This paper is of very high value to the drowning prevention community and the drowning prevention effort!!
Response: Thank you for your feedback, we are pleased you saw value in our work.
Reviewer 2 Report
Comments and Suggestions for Authors
This systematic review, titled "Who’s using ERIC in drowning prevention? Implementation science in drowning prevention interventions in high-income countries," is an excellent contribution to the field of drowning prevention. The comprehensive analysis of implementation science in high-income countries is a valuable resource for researchers, policymakers, and practitioners alike. The meticulous approach to examining ERIC (Expert Recommendations for Implementing Change) in this context is commendable, and the findings presented offer crucial insights into the state of drowning prevention efforts. Overall, this review stands as a valuable reference for advancing evidence-based strategies to reduce drownings in high-income countries.
I recommend that the authors consider incorporating a literary review section in their paper to provide a more comprehensive understanding of the existing research and its relevance to their study. This addition would greatly enrich the depth and context of their work.
I suggest that the authors enhance the conclusion section by summarizing the key findings and their implications more explicitly. Additionally, they might consider discussing potential avenues for future research to further solidify the significance of their work. This would provide a stronger and more comprehensive closure to their study.
Comments on the Quality of English Language
Minor editing of English language required
Author Response
RESPONSE TO REVIEWER 2
Thank you for taking the time to review the manuscript ‘Who’s using ERIC in drowning prevention? Implementation science in drowning prevention interventions in high-income countries: A Systematic Review’ and for your considered feedback. We have responded to your comments and believe this has helped improve the quality of the manuscript.
Comment 1
This systematic review, titled "Who’s using ERIC in drowning prevention? Implementation science in drowning prevention interventions in high-income countries," is an excellent contribution to the field of drowning prevention. The comprehensive analysis of implementation science in high-income countries is a valuable resource for researchers, policymakers, and practitioners alike. The meticulous approach to examining ERIC (Expert Recommendations for Implementing Change) in this context is commendable, and the findings presented offer crucial insights into the state of drowning prevention efforts. Overall, this review stands as a valuable reference for advancing evidence-based strategies to reduce drownings in high-income countries.
Response 1: Thank you for your positive feedback.
Comment 2
I recommend that the authors consider incorporating a literary review section in their paper to provide a more comprehensive understanding of the existing research and its relevance to their study. This addition would greatly enrich the depth and context of their work.
Response 2: Thank you for your comment. Given the current length of the manuscript (64 pages and 18,000 words) and with the main purpose of this manuscript in mind, we have respectfully decided to omit a literary review section, as we believe there are recent papers that better serve this need.
Comment 3
I suggest that the authors enhance the conclusion section by summarizing the key findings and their implications more explicitly. Additionally, they might consider discussing potential avenues for future research to further solidify the significance of their work. This would provide a stronger and more comprehensive closure to their study.
Response 3 Thank you for your feedback. The conclusion has been updated to better reflect the article’s purpose and findings. The change has been highlighted in yellow within the manuscript.
Reviewer 3 Report
Comments and Suggestions for Authors
The article presents a systematic analysis of the application of the Expert Recommendation for Implementing Change (ERIC) concepts and strategies in the context of drowning prevention interventions, using a public health approach as a case study. The article emphasizes that despite the advanced nature of interventions in high-income countries, there is limited information about their implementation. The authors highlight that understanding the process of implementing drowning prevention actions is crucial for improving the design and effectiveness of these interventions.
The article was registered with PROSPERO and followed PRISMA guidelines. The research involved searching eight databases, and the quality assessment was conducted using the Public Health Ontario Meta-tool. Among the 49 included articles, the majority focused on evaluative and iterative strategies, partnership development, and engaging target groups, aligning with the ERIC framework.
Substantively, the article provides valuable information on implementation strategies in drowning prevention, especially in high-income countries. However, despite its rich content, the article loses value due to editorial negligence. Font discrepancies on page 10 affect the readability and aesthetics of the publication. Lack of consistency in content presentation may discourage the reader.
Furthermore, the concluding section should be more specialized, addressing the article's purpose and topic rather than presenting general statements. It would be beneficial for the authors to more precisely relate the conclusions to identified gaps in the literature and suggest specific directions for future research. Conclusions should be an integral part of the article, remaining consistent with its scholarly nature.
In summary, the article is a significant contribution to understanding implementation strategies in drowning prevention. Nevertheless, editorial diligence is necessary to fully extract the potential of the contained content. Corrections in presentation consistency and a more topic-oriented conclusion can significantly enhance the quality of this valuable research work.
Author Response
RESPONSE TO REVIEWER 3
Thank you for taking the time to review the manuscript ‘Who’s using ERIC in drowning prevention? Implementation science in drowning prevention interventions in high-income countries: A Systematic Review’ and for your considered feedback. We have responded to your comments and believe this has helped improve the quality of the manuscript.
Comment 1
The article presents a systematic analysis of the application of the Expert Recommendation for Implementing Change (ERIC) concepts and strategies in the context of drowning prevention interventions, using a public health approach as a case study. The article emphasizes that despite the advanced nature of interventions in high-income countries, there is limited information about their implementation. The authors highlight that understanding the process of implementing drowning prevention actions is crucial for improving the design and effectiveness of these interventions.
The article was registered with PROSPERO and followed PRISMA guidelines. The research involved searching eight databases, and the quality assessment was conducted using the Public Health Ontario Meta-tool. Among the 49 included articles, the majority focused on evaluative and iterative strategies, partnership development, and engaging target groups, aligning with the ERIC framework.
Response 1: Thank-you for your comments.
Comment 2
Substantively, the article provides valuable information on implementation strategies in drowning prevention, especially in high-income countries. However, despite its rich content, the article loses value due to editorial negligence. Font discrepancies on page 10 affect the readability and aesthetics of the publication. Lack of consistency in content presentation may discourage the reader.
Response 2: We have proof-read the paper and corrected font and presentation issues throughout. For example on page 55, line 370 font corrections have been made (see yellow highlighting) and table 4 presentation. We will look to the IJERPH editorial team to provide advice on the editorial issues such as the placement of figure 1, the hanging text on page 10 and as these decisions have been made after submission.
Comment 3
Furthermore, the concluding section should be more specialized, addressing the article's purpose and topic rather than presenting general statements. It would be beneficial for the authors to more precisely relate the conclusions to identified gaps in the literature and suggest specific directions for future research. Conclusions should be an integral part of the article, remaining consistent with its scholarly nature.
Response 3: Thank you for your feedback. The conclusion has been updated to better reflect the article’s purpose and findings. The change has been highlighted in yellow within the manuscript.
Comment 4
In summary, the article is a significant contribution to understanding implementation strategies in drowning prevention. Nevertheless, editorial diligence is necessary to fully extract the potential of the contained content. Corrections in presentation consistency and a more topic-oriented conclusion can significantly enhance the quality of this valuable research work.
Response 4: Many thanks, we have addressed the conclusion issues as noted in comment 3 above. As stated in comment 2, we will look to the IJERPH editors to action the remaining editorial and presentation issues.